

# Microservice security: a systematic literature review

Davide Berardi[1], Saverio Giallorenzo[1,2], Jacopo Mauro[3], Andrea Melis[1], Fabrizio Montesi[3] and Marco Prandini[1]

[1] Department of Computer Science and Engineering, University of Bologna, Bologna, Italy
[2] INRIA, Sophia Antipolis, France
[3] Department of Mathematics and Computer Science, University of Southern Denmark, Odense, Denmark

## ABSTRACT

Microservices is an emerging paradigm for developing distributed systems. With their widespread adoption, more and more work investigated the relation between microservices and security. Alas, the literature on this subject does not form a well-defined *corpus*: it is spread over many venues and composed of contributions mainly addressing specific scenarios or needs. In this work, we conduct a systematic review of the field, gathering 290 relevant publications—at the time of writing, the largest curated dataset on the topic. We analyse our dataset along two lines: (a) quantitatively, through publication metadata, which allows us to chart publication outlets, communities, approaches, and tackled issues; (b) qualitatively, through 20 research questions used to provide an aggregated overview of the literature and to spot gaps left open. We summarise our analyses in the conclusion in the form of a call for action to address the main open challenges.

## INTRODUCTION

Microservices is an emerging development paradigm, where software is built as a composition of multiple services (the "microservices"). Each microservice implements the business logic of a component of the application and is independently executable and deployable. Microservices interact with each other *via* message-passing APIs (*Dragoni et al., 2017*).

Over the last 6 years, microservices have become a popular topic and one of the go-to approaches for many cloud computing projects. According to Web of Science, more than 1,000 articles about microservices have been published since 2014. The year 2020 accounts for more than 400 of them, which points out that interest in the topic is still rising. Microservices are popular because they bring substantial advantages with respect to scalability in cloud environments and flexibility in the process of software development. By separating application components as independent services, software designers can specialise each component by using a dedicated technology and then integrate all such heterogeneous components *via* technology-agnostic APIs.

Corresponding author
Marco Prandini,
marco.prandini@unibo.it

Alas, the advantages of microservices come at a cost: distributed systems are hard to manage, and increasing the number of services of an application gives malicious actors a larger attack surface (*Dragoni et al., 2017*). Several security concerns that are particularly relevant for microservices have been identified by *Chandramouli (2019)*, and early research has already shown that the application of standard patterns for system reliability needs to take new parameters into consideration—like the locations at which the patterns are deployed (*Montesi & Weber, 2018*).

The importance of security in microservices creates the need for understanding and analysing the state of the art for securing this kind of architectures. It is particularly important to understand which problems are especially relevant for microservice systems, and how existing techniques can contribute to addressing them. However, there is still a lack of systematic investigations of studies at the intersection of security and microservice architectures.

Here, we aim to fill that gap by presenting a systematic review of the state of the art of microservice security. We followed a structured approach, which led us to select and gather 290 peer-reviewed publications. At the time of this writing, this constitutes the largest curated dataset on the topic. We first perform a quantitative analysis on the metadata of the publications, for example, publication outlets and keywords. This provides insight into the communities and key research concepts that currently characterise the field. We then map each publication to a vector of 20 different markers, corresponding to 20 research questions on microservices security that we formulated based on established security techniques and the field of microservices as a whole.

Our research questions focused on threat models, security approaches, infrastructure, and development approach. We perform correlation analysis to show that our questions are well-posed (independence), and also to confirm that some topics correlate positively (*e.g.*, Intrusion Detection and Intrusion Prevention, and Agile Development and DevOps as well). Findings from our analysis include: issues with technology transfer from academia to industry on microservices security; lack of guidelines for adopting security by design in microservices; lack of appropriate threat models; lack of guidelines for addressing the attack surface given by technology heterogeneity; and security issues when migrating systems to microservices. Our data, findings, and discussions form a useful basis for orienting future developments of the field.

In summary, the main contributions of this work are:

- the characterisation of Microservices Security as an early-stage, growing research field in need of systematisation and more mature contributions ("Publication Outlets", "Types of Publications");
- the identification of the main research communities on the Microservice Security field and the clustering of authors (Research Communities);
- a presentation of the trends of the main security attacks involving microservice architectures, both from the points of view of threat model (Threat Model) and mitigation (Security Approach (Mitigation));

- a report on the current infrastructural security solutions for microservices (Infrastructure) as well as the interaction between the main microservices development approaches (such as DevOps and Agile) and security (Development);
- a correlation analysis of the answers to our research questions in papers, which sheds light on relationships among the different aspects of microservice security (Correlation between Research Questions);
- a summary of the main open challenges that emerged from our study, which form a call for action for the community of researchers and practitioners working in the field of microservice security (Discussion and Future Directions).

### Structure of the article

We start by providing a summary of related work in "Related Work". In "Review Method" and "Research Questions" we detail the method we followed to conduct the systematic literature review and the research questions, respectively. We present our results in "Review Results" and we conclude in "Discussion and Future Directions" with a discussion on the outstanding challenges.

## RELATED WORK

To the best of our knowledge, the published works that are closest to ours are those by *Vale et al. (2019)* and *Almeida et al. (2017)*. *Vale et al. (2019)* present a systematic mapping that identifies the security mechanisms used in microservice-based systems. Contrary to our work, which provides a general overview on the state of the art of microservices security, the authors narrow their focus on cataloguing the security technologies and mechanisms adopted by developers of microservice-based systems—*e.g.*, authentication and authorisation—leaving out other subjects related to security, like threat models and development methods. Similarly to *Vale et al. (2019)*, *Almeida et al. (2017)* concentrate on surveying the technologies and standards for security, privacy, and communication used in the area of microservice architectures in the cloud.

Extending our view to articles that, at the time of this writing, are not available as peer-reviewed publications, we mention the work by *Hannousse & Yahiouche (2020)* and *Ponce et al. (2021)*. *Hannousse & Yahiouche (2020)* present a systematic categorisation of threats on microservice architectures and propose a selection of possible mitigations. *Ponce et al. (2021)* look at how "security smells" affect microservice-based applications and how to mitigate the effects of such smells through refactoring. As for the proposals by *Vale et al. (2019)* and *Almeida et al. (2017)*, the difference between our work and *Hannousse & Yahiouche (2020)* lies on generality: Hannousse and Yahiouche narrow their investigation down to the threats identified in the literature. Similarly, the work of *Ponce et al. (2021)* focuses on the programming of microservices.

In addition to the related work discussed above, there are quite a few neighbouring surveys with respect to our work that are interesting to discuss: while these studies are not dedicated to the topic of microservice security, they explicitly mention security as an important concern for microservices in different contexts—software engineering, Internet

of Things, containerisation, *etc.* The purpose of reviewing neighbouring related work is twofold:

1. It shows the multifaceted nature of microservice security, giving concrete evidence of the need for an investigation which is both wider and deeper, as we do in this work.
2. It provides a general overview of the challenges and possible uncovered research topics related to security in microservices—which inspired some of the questions presented in "Review Method".

*Dragoni et al. (2017)* present an overview of microservices, including a discussion of the origins of the paradigm, its state of the art, and future challenges. They identify a number of trust and security challenges posed by the paradigm. We mention a few examples. Service reuse, one of the key benefits pushed for in the microservice paradigm, requires adopting secure mechanisms for service authentication and authorisation. The increased granularity and heterogeneity of microservice architectures extends considerably the attack surface of these systems. The sophisticated DevOps infrastructure required to operate microservices effectively is a new attack vector.

*Garriga (2017)* conducted a preliminary analysis towards a taxonomy of microservices architectures. While not addressing in particular security concerns, Garriga reports that the security subject is not extensively addressed, highlighting how monitoring and microservice communication trust chains should receive particular attention.

*Joseph & Chandrasekaran (2019)* reviewed approaches proposed in the literature to deal with the various concerns of microservice-based systems. The authors mention the large attack area offered by microservices subject to insider/privilege-escalation attacks and network security issues.

*Casale et al. (2016)* surveyed the topics of European research projects in the area of software engineering. Regarding microservices security, they highlight four main challenges: increasing the usage of software validation and verification methods; improving the trust and interoperability of services through (self/federated)-certification of outputs based on standards; adopting a security-by-design approach on the whole software lifecycle; and helping developers with addressing discontinuities in the chain of compositionality between services and execution environments—*e.g.*, due to data leakages derived from fragile container-host interactions.

*Lichtenthäler et al. (2019)* investigate and discuss the challenges of migrating monoliths to microservices. They observe that security should be part of the migration planning phase to begin with, and that developers need models and frameworks to help them elicit, track, and manage the (frequently implicit) assumptions and invariants induced by the migration of the legacy system. These observations are shared with *Di Francesco, Malavolta & Lago (2017)*, who suggest that the microservice architectural style has a direct impact on the design of a system and that researchers are still investigating how to leverage its characteristics with respect to system quality and security. *Di Francesco, Malavolta & Lago (2017)* note that there exists uncertainty about the realisation of microservices,

indicating the need for comprehensive references to help programmers in the multifaceted aspects of microservice development.

*Noura, Atiquzzaman & Gaedke (2019)* address the open challenges of interoperability in the Internet of Things (IoT), noting how microservices can constitute a solution for the programming of highly distributed IoT networks and provide two decades worth of research and industrial experience to tackle interoperability in heterogeneous systems. Regarding the general security of IoT systems, *Noura, Atiquzzaman & Gaedke (2019)* note the emergence of security issues (*e.g.*, authentication and access control) when system design permits direct access to resource-constrained devices. Reviewing the many solutions and levels at which IoT interoperability can be tackled, *Noura, Atiquzzaman & Gaedke (2019)* note the challenge of both maintaining and guaranteeing the same level of security when mediating among different technologies.

*Márquez & Astudillo (2019)* examine microservice availability tactics to detect, prevent, mitigate, and recover from faults. They highlight how the tactics for the availability of microservices mainly focus on preventing faults, whereas detection, reaction, and recovery are scarcely addressed. Commenting on related challenges, *Márquez & Astudillo (2019)* report a deficit of solutions to support the restoration of normal functionalities after a microservice architecture suffered from some faults.

*Ahmed et al. (2019)* surveyed robust and flexible service management platforms for IoT systems. Like *Noura, Atiquzzaman & Gaedke (2019)*, they identify microservice architectures as the most suitable architectural pattern to handle the heterogeneity of IoT systems and that the foremost challenge in the field is the robust integration of different technologies. *Ahmed et al. (2019)* also report how conventional security solutions and practices are not suitable to handle the expansion, mobility, resource constraints, and new security requirements of the considered systems.

*Cerny & Donahoo (2016)* investigate service integration from the perspective of separation of concerns and identify problems with conventional service integration design/technologies. They report that the lack of proper cross-cutting concerns in programming technologies make it difficult to capture and guarantee that invariants of a given microservice—specifically, on security—hold when paired with integration components.

*Yang et al. (2014)* survey how cloud computing systems can help scientific research. In their report, they notice how the (micro)service paradigm is useful to make resources available to collaborating researchers by providing a well-defined interface specifying the operations that can be performed on, or with, a given resource. However, they also report that privacy and trust issues are of particular concern to researchers, especially in fields that are processing sensitive data such as medical research. For this, appropriate provenance metadata is required, both to understand how and by whom the data was created and modified, as well as to understand where it has been potentially exposed to corruption. Similar comments are shared also by *Plaza, Daz & Pérez (2018)* in the context of healthcare cyber-physical systems. In particular, proper encryption is reported as a key component for (real-time) data acquisition.

*Soldani, Tamburri & Van Den Heuvel (2018)*, reviewing the "pains and gains" of microservices in the grey literature, found how security generates pains at design-time. Like *Yang et al. (2014)*, *Soldani, Tamburri & Van Den Heuvel (2018)* comment that microservice-based applications should support the consistent determination of the provenance and authenticity of data, noting the paradox of that being in contrast with the heavily-distributed nature of microservice systems. Another (meta) observation by *Soldani, Tamburri & Van Den Heuvel (2018)* is how there is a gap between the industrial understanding and state-of-practice on microservices and the state-of-the-art of academic research, one possible reason being that academics have limited access to industrial-scale microservice-based applications.

*Di Francesco, Lago & Malavolta (2019)* identify, classify, and evaluate the state of the art on architecting with microservices from the perspectives of publication trends, the focus of research, and potential for industrial adoption. On security, they report that it is attracting insufficient research. The works by *Vural, Koyuncu & Guney (2017)* and *Alshuqayran, Ali & Evans (2016)* follow similar modalities and results.

*Bélair, Laniepce & Menaud (2019)* surveyed security of containers, a technology frequently paired with microservices. They report how container security is still in an early phase and it faces unsolved challenges. The results presented by *Bélair, Laniepce & Menaud (2019)* match those by *Sultan, Ahmad & Dimitriou (2019)*, who report the presence of a large number of challenges linked to containerisation because OS kernel sharing introduces security issues absent from virtualisation solutions. *Sultan, Ahmad & Dimitriou (2019)* also highlight the importance of enhancing vulnerability management, digital investigation, and container alternatives.

*Puliafito et al. (2019)* present a survey on the employment of fog computing to support IoT devices and (micro)services. In their study, they report how security is the largest cross-cutting technical concern within critical IoT systems, which necessitates a common baseline and interoperable standards to address security challenges within both hardware and software. In particular, *Puliafito et al. (2019)* advocate for solutions to provide a full-stack secure chain of trust from devices to fog/cloud components, which has been only preliminary explored (as remote attestation techniques). *Trnka, Černý & Stickney (2018)* and *Puliafito et al. (2019)* report also the importance of addressing the concerns of context-aware security (in IoT systems), especially for authentication and authorisation.

Also *Yu et al. (2019)* surveyed the literature on microservice-based fog applications to elicit the security risks threatening them. The main threats highlighted include: kernel-level leakage vulnerabilities linked to containerised deployment; man-in-the-middle/insider attacks on data-transmission interception; the need to verify when services become compromised/misbehave; and network-level vulnerabilities on data-routing alteration.

Table 1, shows the differences between these various works, in numerical and boolean terms. As clearly evincible, our work expands previous work by adding a conspicuous amount of analysed publications; using white literature at its roots and following the trend and methods of the main Systematic White Literature Reviews.

**Table 1 Summary table and comparison with related works.** For each row/work in the table, we report: its reference; its publication year; its type (systematic literature review (SLR), survey, *etc.*); the number of publications it encompasses; whether it analyses white (peer reviewed) literature; whether it analyses grey (blog posts, *etc.*) literature; the sources it used to search its dataset.

| Publication | Year | Type | Num. | White L. | Grey L. | Sources |
|---|---|---|---|---|---|---|
| This work | 2021 | SLR | 290 | ● | ○ | ACM Digital Library |
| | | | | | | IEEE Xplorer |
| | | | | | | SpringerLink |
| | | | | | | Scopus |
| | | | | | | Science Direct |
| | | | | | | Wiley |
| | | | | | | Google Scholar |
| Vale et al. | 2019 | SLR | 26 | ● | ○ | ACM Digital Library |
| | | | | | | IEEE Xplorer |
| | | | | | | SpringerLink |
| | | | | | | Science Direct |
| | | | | | | Wiley |
| | | | | | | Google Scholar |
| Almeida et al. | 2017 | Survey | N.A. | ● | ○ | N.A. |
| Hannousse & Yahiouche | 2020 | SLR | 46 | ● | ○ | ACM Digital Library |
| | | | | | | IEEE Xplorer |
| | | | | | | SpringerLink |
| | | | | | | Science Direct |
| | | | | | | Wiley |
| Soldani et al. | 2018 | SLR | 51 | ○ | ● | Google |
| | | | | | | Bing |
| | | | | | | Duck Duck Go |
| | | | | | | Yahoo! |
| | | | | | | Webopedia |

# REVIEW METHOD

In this section, we describe and motivate the steps we followed to perform our systematic review.

Following the guidelines by *Snyder (2019)*, and as depicted in Fig. 1, we started by searching and retrieving the literature for relevant publications from several data sources by using the same keyword query. We then performed a manual revision process of the automatically selected publications to exclude publications out of the scope of this study and perform snowballing—*i.e.*, recursively adding to the dataset relevant publications cited by the already selected publications. The resulting dataset consists of 290 publications. We analysed these publications to collect statistical and transparent answers to our research questions, which are detailed in "Research Questions".[1]

[1] The list of the publications and their bibliography information is publicly available at https://doi.org/10.5281/zenodo.4774894.

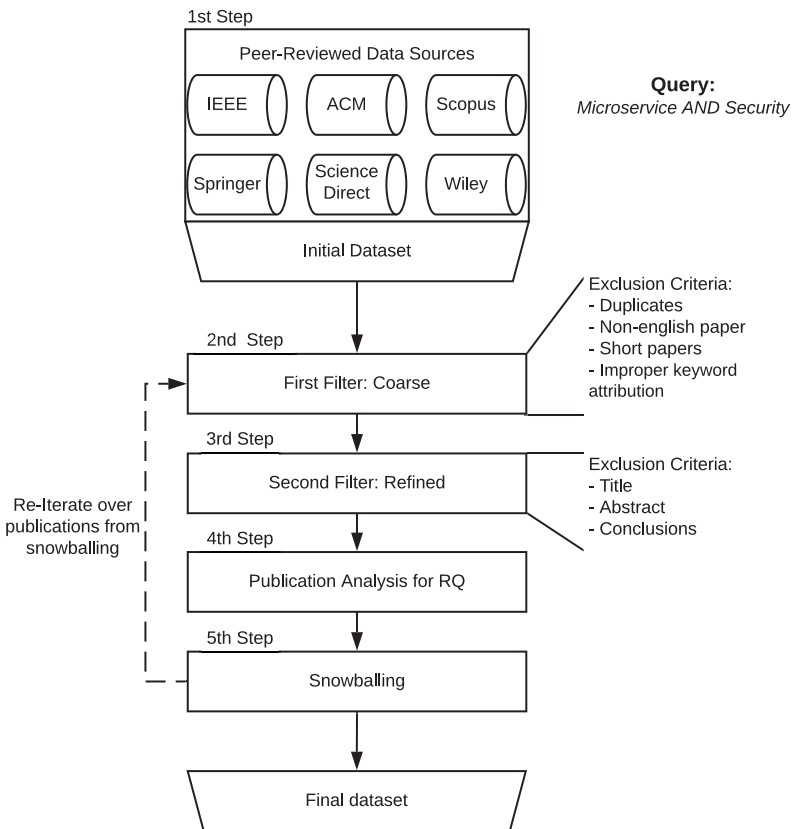

**Figure 1 Schema of the method followed to gather the dataset for this review.**

## Selection query and collection of publications

Security in microservices includes complex and heterogeneous topics, ranging from development to infrastructural concerns. In our choice of a selection query to gather an initial dataset, it was important to pick a sufficiently general query. For this reason, we adopted the query "Microservice AND Security" for our initial search, capturing all the publications containing both terms in any of their title, abstract, or body.[2]

*Di Francesco, Lago & Malavolta (2019)*, *Plaza, Daz & Pérez (2018)*, *Soldani, Tamburri & Van Den Heuvel (2018)* reported how publications on the topic of Microservice started in 2014. Taking into account this fact, we limited our research to contributions published since 2014. During the 7 years covered by our work, the body of knowledge on this topic has grown significantly. For this reason, we deemed it useful to consider white literature only: in terms of quantity, it represents a very meaningful sample of the research produced during the considered time frame, and in terms of quality, it allowed us to rely on peer review. Thanks to the more uniform organisation of white literature, we are also more confident in the level of consistency of our choice and application of the selection criteria. This is not to say that grey literature is not worth investigating. Blog posts, personal websites, technical reports, white papers, *etc.*, are often the preferred venues for practitioners to share ideas. However, as also pointed out in *Soldani (2019)*, "it is very

[2] We performed experiments with potentially more inclusive queries, such as "Microservice AND (Security OR Authorisation)", as well. The tried queries, however, did not extend the search in any useful way since the term "Security" proved to be general enough to cover specialised aspects like authentication, authorisation, and (safe) communication.

difficult to uniquely measure the quality of grey literature when conducting a systematic, controllable, and replicable secondary study" and we are not aware of a standard method for the evaluation of grey literature. Analysing the grey literature was beyond the quality goal of this article and we leave it as future work.

Accordingly to this strategy, we collected publications from 6 different publishers, focusing on peer-reviewed publications. We did not, for example, use Google Scholar or arXiv, since they also list resources that are not peer-reviewed. We list the publishers, reporting the respective numbers of publications that matched our query:

- ACM (https://dl.acm.org/), 478 publications;
- IEEE explore (https://ieeexplore.ieee.org/), 181 publications;
- Springer (https://link.springer.com/), 345 publications;
- Scopus (https://www.scopus.com/home.uri), 134 publications;
- Science Direct (https://www.sciencedirect.com/), 358 publications;
- Wiley (https://onlinelibrary.wiley.com/), 208 publications.

This gave us an initial dataset of 1,704 publications in total. We collected publications published up to the 31st of December 2020, using the academic subscriptions provided by the affiliations of the authors—the University of Bologna and the University of Southern Denmark. To guarantee the same level of trustworthiness and authenticity, we retrieved the publications only from the official entries, avoiding external sources such as the authors' personal websites.

## Publications triage

The publications retrieved from the publishers were processed in three steps to check if they should be excluded according to distinct exclusion criteria. Graphically, in Fig. 1, these steps are labelled as 2nd, 3rd, and 4th Step(s).

In the 2nd Step, we looked at whether the keywords "Microservice" and "Security" were used. We excluded a publication if the keywords appeared only in the bibliography. Moreover, we excluded the publication if it was too short (less than two pages), publications not written in English, and duplicate publications already listed in another publisher source.

In the 3rd Step, we looked at the title, abstract, and conclusion of each publication. Publications that do not treat or discuss topics related to microservices and security were excluded. In this step, we also excluded publications in which the security topic was orthogonal or incidental. In this way, we excluded publications where "microservices and security" was one of the possible application scenarios, but not the main subject of the study. We also excluded cases in which the work tangentially mentioned the satisfaction of some security aspects, without detailing the design/development of the security technologies to accomplish them. For example, we excluded publications focusing on blockchain technologies where the authors incidentally mention authentication and integrity protection as inherent security properties of blockchain-based implementations.

In the 4th Step, we performed an analysis of the publications, answering to the research questions (RQ) detailed in Research Questions. No publications were excluded at this step.

At this point, the following publications remained in the dataset (268 in total):

- ACM, 67 publications;
- IEEE explore, 59 publications;
- Springer, 46 publications;
- Scopus, 28 publications;
- Science Direct, 53 publications;
- Wiley, 15 publications.

### Snowballing

As the last (5th) step for the systematic literature review, we performed a backward snowballing process (*Wohlin, 2014*) with the objective of identifying additional relevant references for our study from the works cited by the already selected publications.

All references collected in this way underwent the triage by following the Steps 2, 3, and 4. Each referenced publication accepted for inclusion by these steps was then added to the dataset of selected publications. Snowballing was recursively performed on these newly-added publications until reaching a fixed point; *i.e.*, until no new publications was added to the dataset.

The outcome of repeatedly applying the snowballing process led to the following results:

- 40 references in the first round, from which we selected 9 publications;
- 22 references in the second round, from which we selected 8 publications;
- 5 references in the third round, from which we selected 5 publications;
- 4 references in the fourth round, where we selected 0 publications.

The 4 cycles of snowballing yielded 22 additional publications that were included in the dataset to reach the final size of 290 publications.

## RESEARCH QUESTIONS

In this section, we detail the research questions that guided our systematic review.

Usually, the research questions for systematic literature reviews are fairly broad and do not amount to more than six. In our case, we chose to adopt more questions (20) but dichotomous (*i.e.*, with yes-or-no answers), to favour precision and objectiveness. To define the questions and seek guidance in categorising the relevant security issues for microservices, we took inspiration from the related work presented "Related Work", as well as from the state of the art in standards and methods, namely the NIST Special Publication 800-204 "Security Strategies for Microservice-based Application Systems" (*Chandramouli, 2019*).

Our questions are collected in four macro groups (**G**s), each covering a different concern.

- **G1:** Threat Model. Questions on threat modelling and how threats are dealt with.
- **G2:** Security Approach. Questions on the security approach, *e.g.*, whether it is preventive, adaptive, proactive, or reactive.
- **G3:** Infrastructure. Questions on the infrastructure that microservices run on.
- **G4:** Development. Questions on the development process.

The questions in each group are reported in the remainder of this section.

### First group: threat model

Mapping the usage of threat models is important to see gaps when a security violation must be handled, or if known models are outdated and need to be adjusted. The NIST report, for instance, hints at the importance of identifying the threats looming over a microservices architecture (*Chandramouli, 2019*). The usage of a formal threat model has proven to be extremely useful in the identification of attack types and their strategic countermeasures (*Death, 2017*).

Several threat models exist in the literature. The most famous one is STRIDE (*Kohnfelder & Garg, 1999*) named after the Spoofing, Tampering, Repudiation, Information Disclosure, Denial of Service, and Elevation of privilege security threats. Other threat models however exists, such as PASTA (*UcedaVelez & Morana, 2015*) or OWASP (*OWASP Foundation, 2020*).

In our review and with this first group of questions, we aimed to understand whether a publication followed a known model, strategy, or guideline. Alternatively, we wanted to know if new security models were proposed.

This group consists of the following questions.

- **Q1:** Does the publication mention STRIDE, or at least consider all of its aspects?
- **Q2:** Even without explicitly mentioning STRIDE, does the publication involve at least one of its aspects (Spoofing, Tampering, …)?
- **Q3:** If STRIDE aspects or equivalent are considered, does the publication propose/discuss a concrete implementation/solution (either developed by the same author or one taken from the literature)?
- **Q4:** Does the publication consider or follow another threat model rather than STRIDE without introducing a new one?
- **Q5:** Does the publication mention policies, workflows, or guidelines to handle violations?

In particular, with question Q1 and Q3 we looked for the adoption of STRIDE, being the most popular threat model. In the remaining questions, we investigate if the publication defined some threat model—either from the literature or a newly one introduced in that publication—or at least discussed equivalent principles or guidelines without mentioning STRIDE.

## Second group: security approach

Many related works cite the usage of preventive measures to secure microservices (*Márquez & Astudillo, 2019*; *Vale et al., 2019*; *Garriga, 2017*; *Almeida et al., 2017*; *Ahmed et al., 2019*; *Soldani, Tamburri & Van Den Heuvel, 2018*) while some indicate the need for further research in the other directions of proaction, reaction, and adaptation (*Vale et al., 2019*; *Márquez & Astudillo, 2019*). With this second block of questions, we wanted to go deeper into the security aspects, considering the specific security approaches, solutions, and also the role that microservices play.

This group consists of the following questions.

- **Q6:** Does the publication mention Intrusion Detection System (IDS) functionalities?
- **Q7:** Does the publication mention Intrusion Prevention Systems (IPS) functionalities?
- **Q8:** Does the publication mention Threat Intelligence?
- **Q9:** Does the publication mention Exfiltration Leaks?
- **Q10:** Does the publication address Insider Threats?
- **Q11:** Are microservices part of the solution?
- **Q12:** Are privacy and GDPR considered?

## Third group: infrastructure

The NIST report by *Chandramouli (2019)* dedicates a large part of its content to infrastructural security solutions for microservices. Similarly, the majority of the mentioned related work in "Related Work" presents or at least cites infrastructural solutions for security, acknowledging that the infrastructure of microservice systems is typically complex, encompassing concerns that span from service deployment and service-to-service coordination (discovery, composition, consistency) to the definition of security-specific mechanisms (authorisation, authentication).

In this group of questions, we aimed at finding information on the infrastructure configurations considered in the publication. This group consists of the following questions.

- **Q13:** Does the publication specify how the proposed architecture is controlled or managed (*e.g.*, in a centralised, decentralised, or hybrid way)?
- **Q14:** Does the publication mention Infrastructure-as-a-Service?
- **Q15:** Does the publication mention service discovery?

## Fourth group: development

Microservices are often associated with software development practices like DevOps and Agile (*Balalaie, Heydarnoori & Jamshidi, 2016*; *Vadapalli, 2018*) which, in turn, are heavily influenced by the inclusion of security-oriented practices (*Casale et al., 2016*; *Lichtenthäler et al., 2019*; *Cerny & Donahoo, 2016*; *Soldani, Tamburri & Van Den Heuvel, 2018*).

In this last set of questions, we aimed at checking the extent to which these practices are used also in the setting of security, for example by verifying whether specific development processes and security standards are considered.

This group consists of the following questions.

- **Q16:** Does the publication mention DevOps, Continuous Integration, Continuous Deployment, or Continuous Delivery?
- **Q17:** Does the publication mention Agile, or how security experts are integrated from a development process point of view?
- **Q18:** Does the publication mention Domain Driven Development?
- **Q19:** Does the publication mention Model Driven Development?
- **Q20:** Does the publication mention certifications, such as ISO27000 (https://www.iso.org/isoiec-27001-information-security.html), or technological standards such as X.509 (https://tools.ietf.org/html/rfc5280)?

## REVIEW RESULTS

In this section, we present the outcome of the literature review. We start by presenting quantitative results from the metadata of the publications in our dataset. This is useful to map the trends over time and current shape of the field, in terms of number of contributions, type (proceedings, articles), communities, and keywords (and their relations). Then, we present qualitative results derived from the analysis of the types of contributions (theoretical, applicative, *etc.*) and of the relation between the selected dataset and our research questions (cf. "Research Questions"). The qualitative part is aimed at providing a detailed insight on existing research patterns, gaps, and uncovered areas of the field. We close the subsection with a correlation analysis of the questions, providing a quantitative look over the relationships between them. For reference, we also report our dataset in tabular form, each entry associated with the positive answers given to our research questions.

### Insights

In the following subsections, we highlight in separate paragraphs (like this one) the main insights that emerge from our analysis. Each insight motivates an open challenge, which we write in bold as the heading of the insight. We will use these challenges in "Discussion and Future Directions" to structure our discussion about useful future directions for research on microservice security.

### Metadata results

We start our quantitative analysis of the collected dataset by presenting in Fig. 2A the time distribution of the selected publications. As expected, security in microservice systems gained a lot of academic interest in the latest years. This is reflected by the sharp increase in the number of publications since 2014. In Fig. 2A, we report the number of collected publications per year.

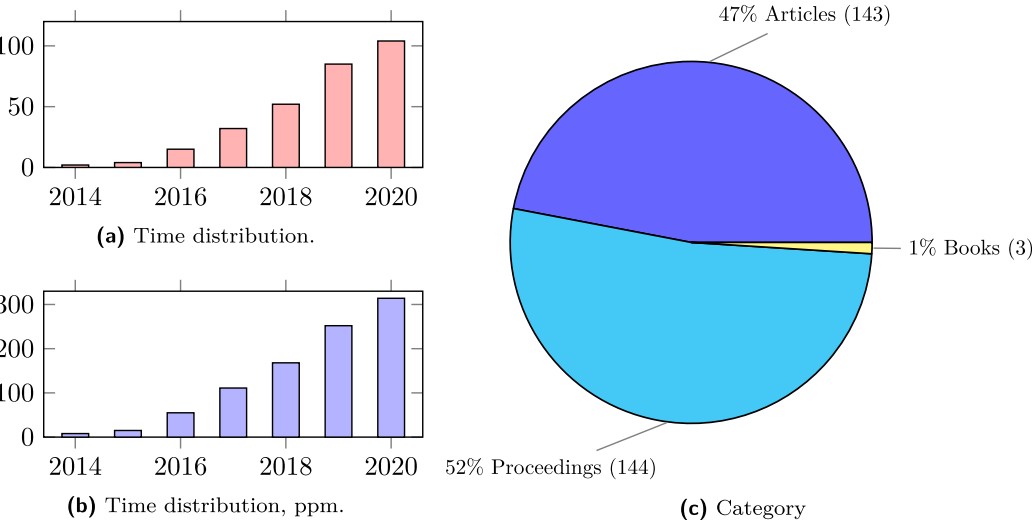

**(a)** Time distribution.

**(b)** Time distribution, ppm.

47% Articles (143)

1% Books (3)

52% Proceedings (144)

**(c)** Category

**Figure 2** **Time and category distribution of publications.**

As a reference to indicate the degree of growth of the field, we report in Fig. 2B the yearly ratio (in parts per million) between the collected publications and the overall number of publications in computer science.[3]

[3] Source: https://dblp.org/statistics/publicationsperyear.html.

### Publication outlets

From the plot in In Fig. 2C, we see that conferences and journal venues are the most commons outlets, while books/collections are underrepresented. This last fact indicates the early stage of the field, where established references are still lacking. However, conference proceedings are almost matched by journal articles, marking a maturing trend of results that are solid enough to constitute material for more structured contributions, as those found in peer-reviewed journals.

We now concentrate on the specific conferences and journals where the publications in our dataset have been published. In Figs. 3 and 4, we report this result in two versions: (i) in tabular form, on the left-hand side of Figs. 3 and 4, with the acronym, the full name, and the number of contributions in our dataset of the venues with the most contributions and (ii) on the right-hand side of Figs. 3 and 4, showing the data on the left as a pie chart.

Regarding the distribution of publications over the different categories of venues, we note how the audience of journals and conferences vary. In fact, there is no predominance of security-oriented or even software engineering venues, which could have been the most likely targets. Instead, the analysed publications appear on publications addressing a broad range of topics, from networking to cloud computing, and on open journals such as IEEE Access and ACM Queue. Furthermore, there is no clear preferred venue that dominates the others, but contributors are rather scattered over many neighbouring venues.

| Acronym | Name | # in dataset |
|---|---|---|
| ARES | International Conference on Availability, Reliability and Security | 2 |
| CCGRID | IEEE/ACM International Symposium on Cluster, Cloud and Internet Computing | 2 |
| EuroS&P | European Symposium on Security and Privacy | 2 |
| ICSA | IEEE International Conference on Software Architecture | 3 |
| IFIP | The International Federation for Information Processing Conference | 3 |
| MEDES | ACM Conference on Management of Digital EcoSystems | 2 |
| NOMS | Network Operations and Management Symposium | 2 |
| SEC | Security Conference | 3 |
| STAF | Software Technologies: Applications and Foundations Interoperable Systems | 2 |

**Figure 3** **Conferences with the largest number of publications in our dataset.**

| Acronym | Name | # in dataset |
|---|---|---|
| CC | Cluster Computing | 4 |
| CCPE | Concurrency and Computation: Practice and Experience | 4 |
| ESE | Empirical Software Engineering | 2 |
| FGCS | Future Generation Computer Systems Conference | 7 |
| FI | Future Internet | 2 |
| IEEE Access | IEEE Access Multidisciplinary open access journal | 5 |
| IEEE IC | IEEE Internet Computing | 3 |
| IEEE PDS | IEEE Transactions on Parallel and Distributed Systems | 3 |
| IST | Information and Software Technology | 2 |
| JSS | Journal of Systems and Software | 8 |
| MNA | Mobile Networks and Applications | 2 |
| MTA | Multimedia Tools and Applications | 2 |
| PCS | Procedia Computer Science | 3 |
| Queue | ACM Queue | 3 |
| SICS | Software-Intensive Cyber-Physical Systems | 2 |
| SPE | Software: Practice and Experience | 4 |
| Sensors | IEEE Sensors Journal | 3 |

**Figure 4** **Journals with the largest number of publications in our dataset.**

We give a twofold interpretation to the phenomenon. On the one hand, this fact can indicate that microservice security is perceived as of cross-disciplinary interest, each contribution seeing it from the lens of its specific area (whether it be software engineering, networks, sensors, cloud computing, *etc.*). On the other hand, we notice the lack of specific venues dedicated to microservices, and least of all, dedicated to microservice security.

## Insights

**Fragmentation of outlets**: there are no reference venues for the area of microservice security (neither journals nor conferences). This makes it difficult for researchers and practitioners to keep up with the state of the art, as well as to find dedicated conventions where they can discuss this topic with the rest of the community interested in the area.

## Research communities

To add more insight on the communities of the field, we also perform a network analysis to identify and explore the clusters of the most prolific authors and their research collaborations. Specifically, we are interested in analysing the networks of collaboration of "core authors", *i.e.*, prolific authors that, by working with different people, act as a liaison among separated groups of authors.

To find the clusters of core authors in our dataset, we consider all the authors in the dataset and we aggregate them in clusters such that each member of a cluster has at least one contribution published with one of the members of the cluster. Since we are interested in "core authors"—*i.e.*, authors with more than 2 works in the dataset—we remove all those clusters formed around just one work—*i.e.*, where the maximum number of publications published by the most prolific author is one.

Our analysis extracted 16 clusters from our dataset. We report in Table 2 the result of our analysis, labelling each cluster from **A** to **P**. For each Cluster, we report the name of the author, the number of publications (# pub.) in our dataset and their affiliation.

The measure gives some interesting insights. First, clusters **F**, **G**, **J**, and **L** are totally localised in one country or the same University/Institute, they are relatively small (compared to the others in the Table), and include some of the most prolific authors (**J** and **L** in particular). Four other clusters follow a different trend: **C**, **H**, **P** and **I**. They are big-size clusters (respectively 6,10,8 and 6), they count one core author (respectively with 3,3,4 and 3 publications) but they are rather homogeneous, the first mainly including authors from Brasil, Finland and the fourth one from Portugal. Clusters **A**, **B**, **D**, **K**, **M**, **N** and **O** are the most varied. Cluster **A**, is the largest (22 authors) and most heterogeneous one: it includes 6 core authors from 5 different countries (Brazil, Germany, Italy, Switzerland, and the UK) and 12 co-authors from 4 countries different from those of the core authors (Australia, France, Portugal and the US). Cluster **B** includes 6 core authors over 24 members, distributed over just 5 countries (Brazil, Germany, Italy, Greece and Switzerland). Cluster **D** includes 8 authors, of which 6 are core and come both from either China or US. Cluster **K** is another big cluster of 16 authors with include 3 core authors from US and Germany. Clusters **M**, **N** and **O** follow the same trend of cluster **D**. This means that these clusters are build around 2 core authors which represent the main affiliation provenance, respectively Holland, Germany and Switzerland, US and UK.

Overall, the communities of core authors in the dataset is distributed among three types of clusters:

- "open" clusters (**A**, **B**, **D**, **K**) of co-authors linked by a few (if not one) core authors and diverse affiliations;

**Table 2 Cluster authors correspondence.**

| Cluster | Author | # Pub. | Affiliation | Cluster | Author | # Pub. | Affiliation |
|---|---|---|---|---|---|---|---|
| A | Fetzer Christof | 3 | TU Dresden | G | Makitalo Niko | 1 | University of Helsinki |
| A | Brito Andrey | 2 | Universidadede Campina Grande | H | Jin Yike | 1 | Unknown affiliation |
| A | Kopsell Stefan | 2 | TU Dresden | H | Yu Dongjin | 1 | Hangzhou Dianzi University |
| A | Pietzuch Peter | 2 | Imperial College London | H | Zhang Yuqun | 1 | Southern University |
| A | Pasin Marcelo | 2 | University de Neuchâtel | H | Zheng Xi | 3 | Xi'an Jiaotong University |
| A | Felber Pascal | 2 | University of Neuchâtel | H | Zhang Chong | 2 | Chong Qing Hospital |
| A | Fonseca Keiko | 1 | Universidade do Paraná | H | Liu Xiao | 2 | Tsinghua University |
| A | Rosa Marcelo | 1 | University of Melbourne | H | Li Rui | 2 | Facebook |
| A | Gomes Luiz | 1 | Arizona State University | H | Liu Huai | 2 | Universiy of Washington |
| A | Riella Rodrigo | 1 | Universidade do Paraná | I | Donahoo Michael J | 2 | Carnegie University |
| A | da Silva MS Leite | 1 | Universidade Campina Grande | I | Cerny Tomas | 6 | Baylor University |
| A | de Oliveira SV Fernando | 1 | Universidade de Campina Grande | I | Sedlisky Filip | 1 | University In Prague |
|  | Kelbert Florian | 1 | Elastic | I | Walker Andrew | 2 | Carnegie University |
| A | Gregor Franz | 1 | TU Dresden | I | Svacina Jan | 2 | Baylor University |
| A | Pires Rafael | 1 | University of Sao Paulo | I | Bushong Vincent | 2 | Baylor University |
| A | Schiavoni Valerio | 1 | University of Neuchâtel | I | Bures Miroslav | 2 | University In Prague |
| A | Mazzeo Giovanni | 2 | MDM-IMM-CNR lab | I | Tisnovsky Pavel | 2 | University In Prague |
| A | Oliver John | 1 | UC Berkeley | I | Frajtak Karel | 2 | University in Prague |
| A | Romano Luigi | 1 | Universita della Campania | I | Shin Dongwan | 2 | Korea Institute of Energy Research |
| A | Brenner Stefan | 1 | TU Braunschweig | I | Huang Jun | 2 | Duke University |
| A | Hundt Tobias | 1 | UCL Institute of Child Health | J | Yarygina Tetiana | 4 | University of Bergen |
| A | Kapitza Rudiger | 1 | TU Braunschweig | J | Otterstad Christian | 3 | University of Oslo |
| B | Artac | 1 | Necmettin Erbakan University | J | Lysne Olav | 1 | Simula Research Laboratory |
| B | Casale Giuliano | 2 | Imperial College London | J | Hole Kjell J | 1 | Simula Research Laboratory |
| B | Van Den Heuvel W-J | 2 | Tilburg University | J | Ytrehus | 1 | University of Tromso |
| B | van Hoorn Andre | 5 | University of Stuttgart | J | Aarseth Raymond | 1 | University of Tromso |
| B | Jakovits Pelle | 1 | University of Tartu | J | Tellnes Jorgen | 1 | University of Bergen |
| B | Leymann Frank | 1 | University of Stuttgart | J | Bagge Anya Helene | 1 | University of Bergen |
| B | Long Madeleine | 1 | University of Oslo | K | Cecconi Alessio | 1 | Vienna University |
| B | Papanikolaou Vicky | 1 | National School of Public Health | K | Di Ciccio Claudio | 1 | Sapienza University of Rome |
| B | Presenza Domenico | 1 | University of Rome | K | Dumas Marlon | 1 | University of Tartu |
| B | Russo Alessandra | 1 | University of Catania | K | Garcia-Banuelos Luciano | 1 | Tecnologico de Monterrey |
| B | Chesta Cristina | 1 | University of Chester | K | Lopez-Pintado Orlenys | 1 | University of Tartu |
| B | Di Nitto Elisabetta | 1 | Politecnico di Milano | K | Lu Qinghua | 3 | Universtiy of delaware |
| B | Gouvas Panagiotis | 2 | University of Athens | K | Mendling Jan | 1 | Humboldt-Universität zu Berlin |

(Continued)

| Cluster | Author | # Pub. | Affiliation | Cluster | Author | # Pub. | Affiliation |
|---|---|---|---|---|---|---|---|
| B | Stankovski Vlado | 2 | University of Ljubljana | K | Tran An Binh | 1 | CSIRO |
| B | Symeonidis Andreas | 1 | University of Thessaloniki | K | Weber Ingo | 3 | TU Berlin |
| B | Zafeiropoulos Anastasios | 2 | University of Athens | K | Binh Tran An | 2 | CSIRO |
| B | Soldani Jacopo | 1 | University of Pisa | K | O'Connor Hugo | 2 | CSIRO |
| B | Avritzer Alberto | 4 | eSulabSolutions | K | Rimba Paul | 2 | CSIRO |
| B | Ferme Vincenzo | 3 | Kiratech S.p.A. | K | Xu Xiwei | 2 | National Institute of Natural Hazards |
| B | Janes Andrea | 3 | The James Hutton Institute | K | Staples Mark | 2 | CSIRO |
| B | Russo Barbara | 3 | Free University of Bozen-Bolzano | K | Zhu Liming | 3 | CSIRO |
| B | Schulz Henning | 3 | Novatec Consulting GmbH | K | Jeffery Ross | 2 | Mayo Clinic |
| B | Menasche | 3 | University of Rio de Janeiro | L | Mirri Silvia | 2 | University of Bologna |
| B | Rufino Vilc | 3 | UFRJ | L | Melis Andrea | 4 | University of Bologna |
| B | Trubiani Catia | 1 | Gran Sasso Science Institute | L | Prandi Catia | 2 | University of Bologna |
| B | Bran Alexander | 1 | University of Exeter | L | Prandini Marco | 4 | University of Bologna |
| C | Rocha Carla | 1 | Rutgers University | L | Salomoni Paola | 2 | University of Bologna |
| C | Leite Leonardo | 3 | University of São Paulo | L | Callegati Franco | 3 | University of Bologna |
| C | Kon Fabio | 3 | University of São Paulo | L | Giallorenzo Saverio | 2 | University of Bologna |
| C | Milojicic Dejan | 1 | Hewlett Packard Labs | L | Delnevo Giovanni | 1 | University of Bologna |
| C | Meirelles Paulo | 3 | University of São Paulo | L | Monti Lorenzo | 1 | University of Bologna |
| C | Pinto Gustavo | 2 | University of São Paulo | M | Panichella Annibale | 4 | Delft University of Technology |
| D | Hou Kaiyu | 3 | Northwestern University | M | Jan Sadeeq | 1 | Technology Peshawar Pakistan |
| D | Wu Xiaochun | 3 | Zhejiang University | M | Arcuri Andrea | 1 | Kristiania University College |
| D | Leng Xue | 3 | Zhejiang University | M | Briand Lionel | 1 | University of Ottawa |
| D | Li Xing | 3 | University of Chicago | M | Olsthoorn Mitchell | 2 | Delft University of Technology |
| D | Yu YinBo | 1 | Wuhan University | M | van Deursen Arie | 2 | Delft University of Technology |
| D | Wu Bo | 3 | Google Inc. | N | Zimmermann Olaf | 5 | HSR University of Rapperswil |
| D | Chen Yan | 3 | Lunghwa University | N | Stocker Mirko | 1 | HSR University of Rapperswil |
| D | Yu Yinbo | 2 | Wuhan University | N | Zdun Uwe | 3 | University of Vienna |
| E | Nikouei Seyed Yahya | 3 | Binghamton University | N | Lubke Daniel | 1 | Leibniz Universität Hannover |
| E | Xu Ronghua | 2 | Binghamton University | N | Pautasso Cesare | 1 | University of Lugano |
| E | Chen Yu | 3 | University of Singapore | N | Kapferer Stefan | 2 | Witten/Herdecke University |
| E | Blasch Erik | 2 | Air Force Research Lab | N | Wittern Erik | 2 | Witten/Herdecke University |
| E | Aved Alexander | 2 | US Air Force Research Lab | N | Leitner Philipp | 2 | University of Gothenburg |
| E | Nagothu Deeraj | 1 | Binghamton University | O | Michalas Antonis | 1 | Tampere University of Technology |
| E | Faughnan Timothy R | 1 | Binghamton University | O | Paladi Nicolae | 1 | Research Institutes of Sweden |
| F | Sukaridhoto Sritrusta | 3 | Politeknik Surabaya | O | Dang Hai-Van | 3 | University of Westminster |
| F | Panduman YY Fridelin | 1 | Politeknik Surabaya | O | DesLauriers James | 2 | CNRS |
| F | Tjahjono Anang | 1 | Politeknik Surabaya | O | Kiss Tamas | 2 | CNRS |

| Cluster | Author | # Pub. | Affiliation | Cluster | Author | # Pub. | Affiliation |
|---|---|---|---|---|---|---|---|
| F | Falah Muhammad Fajrul | 2 | Politeknik Surabaya | O | Ariyattu Resmi C | 2 | Carleton University |
| F | Al Rasyid MU Harun | 2 | Politeknik Surabaya | O | Ullah Amjad | 2 | Carleton University |
| F | Wicaksono Hendro | 2 | Politeknik Surabaya | O | Bowden James | 2 | Carleton University |
| G | Kilamo Terhi | 1 | Aalto University | O | Krefting Dagmar | 2 | HTW Berlin |
| G | Lwakatare Lucy Ellen | 1 | University of Helsinki | O | Pierantoni Gabriele | 2 | University of Westminster |
| G | Karvonen Teemu | 1 | University of Helsinki | O | Terstyanszky Gabor | 2 | University of Westminster |
| G | Heikkila | 1 | University of Oulu | P | Basso Tania | 1 | Universidade Estadual de Campinas |
| G | Itkonen Juha | 1 | Aalto University | P | Antunes Nuno | 3 | University of Coimbra |
| G | Kuvaja Pasi | 1 | Aalto University | P | Vieira Marco | 1 | University of Coimbra |
| G | Mikkonen Tommi | 2 | University of Helsinki | P | Santos Walter | 1 | Universidade Estadual de Montes Claros |
| G | Oivo Markku | 1 | University of Oulu | P | Meira Wagner | 1 | Universidade Federal de Minas Gerais |
| G | Lassenius Casper | 1 | Aalto University | P | Flora Jose | 4 | University of South Carolina |
| G | Kalske Miika | 1 | University of Helsinki | P | Goncalves Paulo | 2 | Universidade de São Paulo |

- "semi-open" clusters (**C**, **G**, **M**, **N** and **O**) of localised collaborators with sporadic, external collaborations;
- "closed", localised clusters (**F**, **L**, **P**) that tend to be small but whose core authors tend to be the most prolific (**L**).

Given their larger reach, semi-open and open clusters have a better chance to gather an impactful community around the topic. Our call to the authors in the field (particularly the closed clusters that tend to be prolific but rather localised) is to establish international collaborations and coordinate to foster the advancement and growth of the field.

## Concepts and keywords

We conclude our quantitative analysis by providing a graphical representation of the main keywords present in the abstract of the contributions in our dataset. To conduct our analysis, we used VOSviewer by *Van Eck & Waltman (2010)*, a software that offers text mining functionalities for constructing and visualising co-occurrence networks of important terms extracted from a given *corpus*. Specifically, we ignored basic words and copyright statements, and performed a full-count of the words present in the text. We considered only words occurring more than 15 times, sizing them by their relevance in terms of occurrences. The resulting graph, however, is still too large and dispersive to convey useful information: for the sake of clarity we present here a visualisation including only the top 60% most-occurring words.

We report the visualisation of the analysis in Fig. 5.

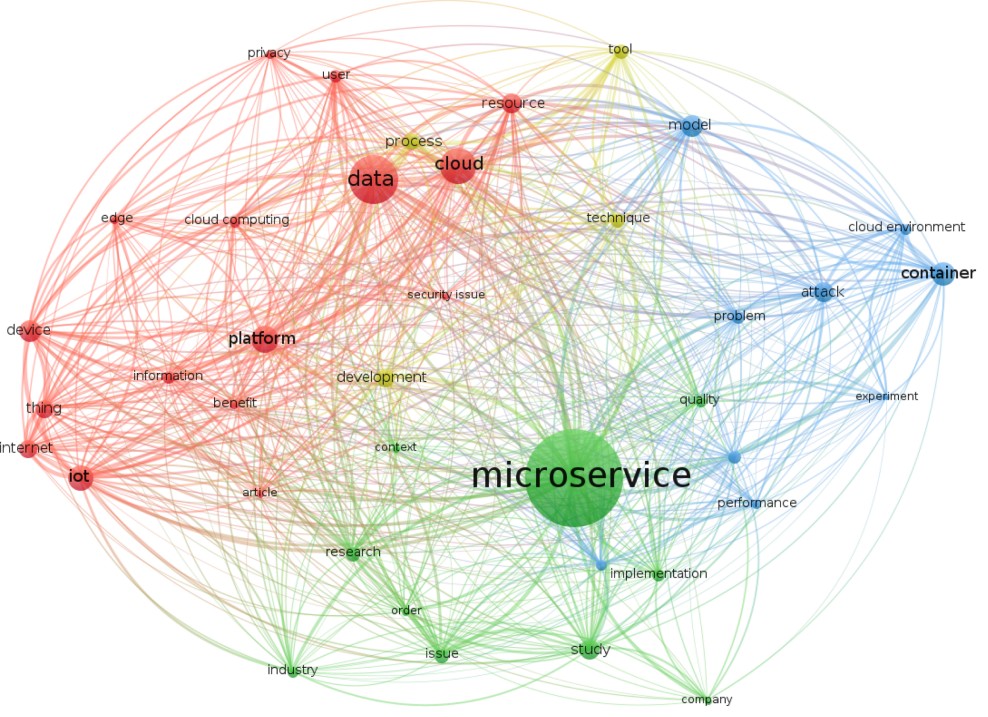

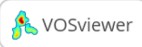

**Figure 5 Word-Net of the abstracts in our dataset.**

VOSviewer automatically clustered the words in 4 areas using its modularity-based clustering algorithm, which is a variant of the cluster algorithm developed by *Clauset, Newman & Moore, 2004* to detect communities (clusters) in a network that also considers modularity.

We can interpret the clusters as follows:

- The blue area marks the main terms of this study, grouping words like *microservice* and *system*. The result does not surprise, since those words describe the design of the systematic selection we performed.
- The green area marks technical terms as *container* or *attack*.
- The red area identifies application terms, *e.g.*, the targets or reasons of the research, if it is an industrial or research-focused article. We find for instance the word *Internet-of-Things*, as it is mainly cited with industry and research applications rather than along with terms like *container* and *cloud*.
- The yellow area includes words that identify the subject of a study, whether it be some *tool*, *data* (of the system, of the users), *user*s, and they *privacy*. The word *tool* here is peculiar, as it acts as a bridge between the other areas. Also this finding is somehow expected, as the field of microservice security is marked by a fairly practical orientation towards automatisation of processes and control.

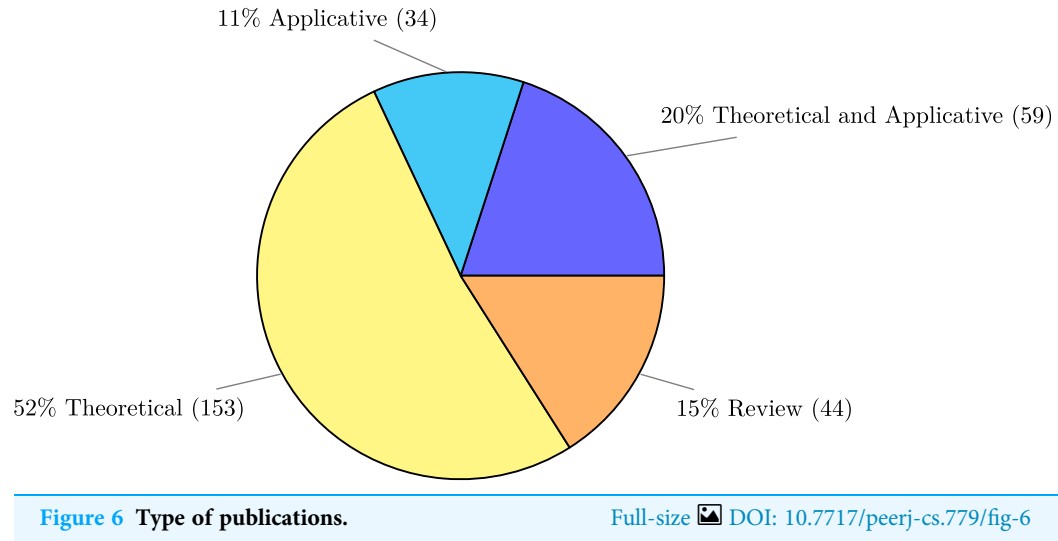

**Figure 6** Type of publications. 

## Publication context analysis

In this section, we discuss trends and considerations derived from reading the selected publications and the research question detailed in "Research Questions".

### *Types of publications*

In Fig. 6 we report the distribution of the type of research contribution—whether theoretical, practical, mixed, or a review.

More precisely, regarding the type of research contribution, we mapped every publication in our dataset to one of the following types:

- *Theoretical* for publications that present an approach for a specific problem without any implementation artefact.
- *Applicative* for publications that describe an implemented application possibly with its validation.
- *Theoretical and Applicative* for publications that develop a theory and provide a practical tool, framework, program, or application.
- *Review* for both literature reviews and social studies (*e.g.*, on developers).

Reviews constitute the 15% of works, marking the fragmented shape of the field, which is in rapid expansion and in need for studies to map its research landscape. Besides surveys, the other contributions in the field are distributed among a 52% share that introduce new theoretical results, a 20% share that contribute by pairing new theoretical proposals with implementations, and the remaining 11% describing pure applications. The fact that the main publications in the field are of theoretical nature is surprising, given the prominently applied nature of microservices. Indeed, excluding surveys, we have that for every 5 publications slightly more than 3 (64% of them) are of purely theoretical. We attribute this figure to two phenomena. The first marks the current exploratory trend of the field, which is still engaged in proposing new ideas and in evaluating and maturing them into models amenable to implementation. The second phenomenon relates to the impact

that microservices have at the processes/organisational level, with works that are intrinsically theoretical because their contribution can be hardly crystallised into automated implementations, *e.g.*, for proposals of attack models or techniques for handling security within organisations and development teams notwithstanding the possible explanations above, it is worth noting the (quantitative) distance between contributions from academia and applications available to practitioners and the industry, which is an indicator of an untapped potential for joint synergies between the two communities.

After having characterised the type of publications in the field, we proceed by exploring the results from the answer of the research questions following the 4 macro-groups presented in "Research Questions".

## Insights

**Technology transfer**: the field of microservice security is still in the early phase of new idea proposals. There are just a few implementations of these ideas, which hinders industrial adoption.

### Threat model

A total of 176 publications (ca. 65% of the dataset) give a positive answer to at least one question of this category. However, only 53 publications among those 120 (ca. 30% of the total dataset) mentioned the usage of at least one threat model to analyse or classify threats The reason for those publications to adopt a threat model vary, from publications that use the model to motivate their proposed solutions to reviews that use the model to structure their overview of the state of the art. Interestingly, in ca. 80% of those publications that mention the usage of at least one known threat model, the model is tailored to work on a specific application scenario. This is an indication of the lack of usage of a generic threat model for microservice security. We conjecture that this lack of usage of generic threat models is due to the fact that the majority of research done on microservice security comes from the software (engineering, languages) side of the field, rather than from the side of security, which advocates for a security-by-design approach.

A complementary explanation of that phenomenon is that there is no affirmed threat model for microservices, *e.g.*, due to the difficulty of making the model specific enough for microservices yet avoiding the infamous problem of threat explosion, where the effort required to prioritise and consider all threats starts exceeding the benefits of proposing methods to manage them *Wuyts et al. (2018)*. Threat explosion is a known problem of neighbouring areas to microservices, like cloud, edge, and fog computing (*Di Francesco, Malavolta & Lago, 2017*; *Ibrahim, Bozhinoski & Pretschner, 2019*; *Guija & Siddiqui, 2018*; *Lou et al., 2020*; *Flora, 2020*; *Truong & Klein, 2020*; *Russinovich et al., 2021*) where the authors resorted to defining smaller, customised threat models rather than adopting standard ones, due to the problem of requiring conspicuous adaptation efforts to tailor them to such complex and multifaceted architectures.

Regarding the possible attacks addressed in the publications, Fig. 7 categorises the publications based on the STRIDE threats, following up on question Q2 asking if the

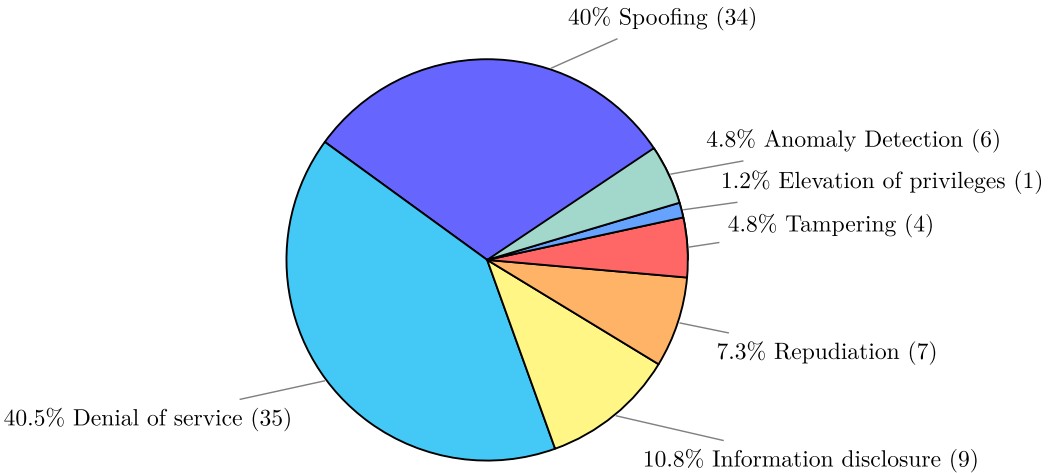

**Figure 7** **Attack type identified following the STRIDE classification.**

publication involves at least one of threats of the STRIDE classification. The most commonly tackled attacks are of the "spoofing" and "denial of service" kinds. This is an effect of the push for fine-granularity and independence of services advocated by microservices, where applications result from several small (in size), independent software components that communicate with each other. Such decentralised communication/coordination is one of the most important attack vectors for microservice applications, in particular, the possibility to disguise a communication from an unknown source as being from a known, trusted source, which matches the spoofing attack category. Such attacks, along with tampering and repudiation ones (which together represent more then half of the attack types found in our collection), entail the need for solutions to address attacks centred around exploits of data provenance.

A similar consideration can be made for denial-of-service attacks, where the flexible scalability of microservices allows malicious intruders to, *e.g.*, scale up peripheral microservices and hit more central and well-protected components with (distributed) overpowering attacks.

## Insights

**Adoption of security-by-design**: security in microservices frequently comes as an afterthought, whereas it should be one of the main concerns for their engineering.

**Data provenance**: the quantity of spoofing, tampering, and repudiation attacks highlights the need to address the general problem of data provenance in microservices.

**Dedicated attack trees and threat models**: while there are attacks that specifically pertain to microservices, such as those that leverage the scalability of microservice architectures to cause denial of service, there are no dedicated threat models to help developers become aware of those particular threats.

*Security approach (Mitigation)*

In terms of solutions to security issues proposed by the publications (questions Q6–Q10), the most common approach (45 publications) is to address specific problems, such as authentication or exfiltration, rather than suggesting a general approach. Publications dealing with architectural aspects rarely address the overall picture (only 25, roughly 8%, publications focuses on IDS, IPS, Exfiltration Leaks and Threat Intelligence). Again, they focus on local threats like intra-communications or authentication (question Q11). These observations suggest that there is a lack of security approaches that address applications across the full stack.

As far as privacy and GDPR are involved (question Q12), surprisingly, only 9 publications consider privacy protection relevant or worthy of analysis. In particular, only one publication *Badii et al. (2019)* considers the GDPR as a guideline to follow in order to protect the privacy of users. Example of this kind of guideline application are shown in *Voigt & Von dem Bussche (2017)*. Considering that many of the solutions included in the dataset are Cloud-based solutions, it is surprising to note that only one publication claims to be GDPR compliant.

## Insights

**Global view/control**: the distributed nature of microservices introduces the need for technologies that provide global yet decentralised observability and control, *i.e.*, tools that aid in the enforcement of security policies over a whole architecture without single points of failure.

**React & recover techniques**: while we found solutions to prevent and detect attacks, there are only a few proposals about how microservice systems could react to and recover from them.

**Comprehensive technological references**: microservices use diverse sets of technology stacks, each characterised by peculiar exploits. To secure microservice architectures effectively, implementors need dedicated technological references to avoid known threats.

*Infrastructure*

We start the discussion by first focusing on the type of microservice infrastructure used by the various contributions. Specifically, we have 205 publication in our dataset that answer positively to question **Q13**. The breakdown of the answers is:

- 39% (80) describe a centralised approach;
- 24% (49) use a decentralised approach;
- 17% (35) resort to a hybrid approach;
- 20% (41) do not specify which approach they use.

The most widely adopted turns out to be the centralised one. We conjecture two explanations behind this observation. First, the centralised approach has the merit of simplifying the definition, deployment, monitoring, and evolution of policies holding over all the components in a given architecture—traded off with scalability issues and single-point-of-failure concerns. Second, we note that, among the approaches that appeared early

in the literature, many focused on converting monolithic applications into microservice applications. Clearly, having a centralised controller that manages the orchestration of microservices helps this process and is closer in spirit to the monolithic workflow. However, the advent of federated, multi-cloud solutions (that prevent the identification/deployment of a centralised authority over the whole peer network) as well as new distributed-consensus technologies (*e.g.*, blockchains), has led to a decentralisation of control, making new decentralised or hybrid solutions emerge (in our dataset) starting from 2018. As an example, in 2015 and 2016, we find publications such as *Callegati et al. (2016)* and *Lysne et al. (2016)* which presented centralised approaches to enable security in microservice platforms, while starting from 2018 hybrid and decentralised solutions appear like *Pahl & Donini (2018)* for certificate-based authentication or *Andersen et al. (2018)*, *Andersen et al. (2017)* where authors propose a decentralise high-fidelity city-scale emulation to verify the scalability of the authorisation tier.

We notice that the advent of new distributed-consensus technologies also affected the orchestration approach of microservice solutions. For example, works such as *Xu et al. (2019)* propose a decentralised, blockchain-based data-access control for microservices. Recent contributions also tackled the problem of authentication and authorisation in decentralised settings, *e.g.*, *Bánáti et al. (2018)* develops a workflow-oriented authorisation framework to enforce authorisation policies in a decentralised manner, *Taha, Talhi & Ould-Slimanec, 2019* presents a new algorithm that distributes tasks on clusters of vehicular ad-hoc networks, *Zhiyi, Shahidehpour & Xuan, 2018* proposes a secure decentralised energy management framework, and *Tourani et al. (2019)* describes a decentralised data-centric SECurity-as-a-Service (SECaaS) framework for elastic deployment and provisioning of security services. Another interesting work has been done in *Falah et al. (2020)* where authors brought the concept of a digital twin to show how a microservice infrastructure approach can speed up the process of deploying complex infrastructure components.

Infrastructure as a Service (IaaS), which is the focus of question Q14, is also a recurrent topic in our dataset, with 66 publications yielding a positive answer. IaaS include solutions that provide and manage low-level infrastructural components, like computing resources, data storage, network components, *etc.* We notice that IaaS is mentioned mainly as the modality used to deploy the solution but is not studied as a security subject/mechanism *per se*. Works such as *Sultan, Ahmad & Dimitriou, 2019* emerge as exceptions; their authors analysed the security benefits obtained using a container-based infrastructure exposed as a service.

Question Q15 investigates Service Discovery, *i.e.*, the automatic detection of services and their functionalities available in a given architecture/network. A total of 16 publications mention Service Discovery in the context of security. Mainly, they propose architectures that support reactive mechanisms for the detection of security issues. Of those, only 2 mention service registration procedures that include data for performing the preventive analysis of the composition, with the goal of statically finding and fixing possible vulnerabilities and misconfigurations: *Callegati et al. (2018)* and *Kamble & Sinha (2016)*.

### Insights

**Global view/control**: while there is not a definitive approach to microservice security control (whether it be centralised, decentralised, or hybrid), there is a recognised need for applying security control policies in a consistent way across all microservices belonging in the same architecture.

### Development

DevOps and Agile are recurring topics in our dataset. Based on the answer to question Q16, 76 publications used the DevOps approach, while, answering to Q17, 57 used Agile methods—of those 99 publications which represent the 40% of all publications in our dataset, 10 mention both approaches. There is a common consensus in these publications that Agile/DevOps is important in security because microservices seem to be the perfect match for this type of software development model (*Vehent, 2018*; *Hsu, 2018*). In particular, microservices align with the tenet of both approaches: to assign dedicated, independent teams to the development of small and independent components within the architecture Continuous Integration (CI) process. However, the majority of the selected publications provide no in-depth security analysis of any of the two development approaches, but rather indicate the inclusion of generic security measures in the steps of the development method. Only three works, namely *Mansfield-Devine (2018)*, *Anisetti et al. (2019)* and *Kumar & Goyal (2020)*, propose concrete and specific variants of the DevOps methodology that tackle security issues—in particular *Mansfield-Devine (2018)* explicitly cites the guidelines of DevSecOps *Hsu (2018)*.

Migration is one of the main challenges faced in this context; migrating applications introduces important security concerns (*Lwakatare et al., 2019*) that are difficult to track, due to the lack of appropriate devices (both organisational and linguistic) to elicit them from the source codebase and make sure they hold in the migrated one. Another major challenge is the coordination between development teams in the context of privacy-handling issues (*Gupta, Venkatachalapathy & Jeberla, 2019*). Also, security becomes a challenging aspect since the (small, independent) teams need to know many aspects of security (*Leite et al., 2019*) and those DevOps criteria for testing, building, and deployment automation are often neither properly followed in industrial environments (*Bogner et al., 2019*), nor for automated scans (*Chondamrongkul, Sun & Warren, 2020*).

When considering domain- and model-driven approaches (questions Q18 and Q19), 16 publications consider domain-driven approaches and 26 consider model-driven ones, such as *Kapferer & Zimmermann (2020)*, *Avritzer et al. (2020)*. These topics are therefore not as widespread as DevOps. Moreover, all citations in these cases are just brief references of the development approach, and lack a discussion on how one of the two approaches can be used in a security context on microservices.

The last question in this category, Q20, concerns security standards, *i.e.*, curated sets of technologies, policies, concepts, safeguards, guidelines, assessments, procedures, training programmes that should be adopted to reduce security risks and mitigate attacks. The answers we gathered for this question surprised us. Indeed, security standards are a staple

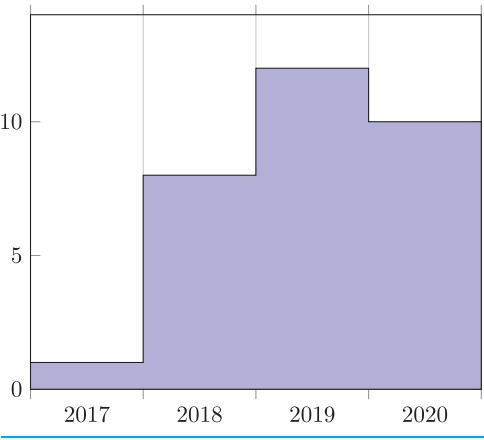

**Figure 8  Blockchain trend.**               

element of industries and organisations that want to impose and guarantee a certain level of security on their members and collaborators (often also for certification purposes – *Stewart, Chapple & Gibson, 2012*, *Lie, Sánchez-Gordón & Colomo-Palacios, 2020*). Despite their widespread use in practice, only 7 publications mention security standards.

In particular, *Souppaya, Morello & Scarfone (2017)* mentions the usage of X.509 to verify a secure method for key exchange between microservice. In *Brenner et al. (2017)* the authors show a solution for securing microservices through the SGX Intel Standard. The authors of *Vassilakis, Panaousis & Mouratidis (2016)* analyse the concept of Small-Cell-as-a-Service, *i.e.*, a technological paradigm for the development of Virtualised Mobile Edge Computing Environments, using several mobile standards for 5G and SDN networks (*e.g.*, MobileFlow *Pentikousis, Wang & Hu, 2013* and VNFs *Agarwal et al. (2019)*). Finally, *Yarygina (2018)* performs a deep analysis on securing microservices, citing and analysing several know standards for both microservice management and security purposes.

## Insights

**Migration to microservices**: there are no established techniques to help developers migrate legacy systems to microservice architectures, and in particular to identify the possible security threats that come from such a migration.

**DevSecOps**: Agile and DevOps practices are widely used when developing microservices, yet only a few publications address how security is addressed and combined in these practices.

### Additional considerations

By analysing our dataset, we were surprised to find many citations to blockchain technologies (as reported above) as well as the lack of more and more mainstream technologies like service mesh and serverless.

Regarding blockchain technologies, we found 31 publications mentioning or explicitly using blockchains. The decentralisation and independence of microservices constitute a

good pairing for the usage of blockchain technologies. Figure 8 presents also the trend of publications using blockchain in the dataset. There is an increasing interest in blockchain applications for microservice architecture. Examples of that pairing include works such as *Nagothu et al. (2018)* and *Xu et al. (2019)*, where the trust-chain of the blockchain is combined with a decentralised microservice architecture to create strong smart contract systems, or *Lu et al. (2021)*, where authors proposed a model-driven engineering approach for blockchain applications with microservice.

New approaches for microservices design and usage such as service mesh *Li et al. (2019a)*, *i.e.*, a dedicated infrastructure layer for facilitating service-to-service communications between microservices is just mentioned by 3 works: *Pahl & Donini (2018)*, where the authors indicate a service mesh architecture for authenticating services—securely adding information to their executables and validating the correct execution of distributed entities with such certificate-based approach—and *Suneja, Kanso & Isci (2019)*, which mentions the service-mesh sidecar pattern used to control security. Another interesting work regarding service mesh is *Hahn, Davidson & Bardas (2020)* where authors analysed under several scenarios issues and challenges in Service Meshes.

Similarly, serverless *Hendrickson et al. (2016)* is mentioned only in 4 publications. We did not expect to find (50%) more citations of serverless than those regarding service mesh. Serverless is a cloud computing execution model in which the cloud provider dynamically manages the allocation/scaling of machine resources depending on inbound requests. Indeed, while the service mesh is a technology from the (micro)service-oriented context, serverless is a more neighbouring concept to that of stateless microservice deployment.

In this context, the most relevant publication is *Casale et al. (2019)*, which presents the results of a European research project to develop a model-driven DevOps framework for creating and managing applications based on serverless computing. Its main result consists in designing applications as fine-grained and independent microservices that can efficiently and optimally exploit the serverless paradigm. The serverless term, despite starting to get momentum, is still loosely related to microservices.

Given their increasing importance and impact in the industry and their close relation with microservices, we argue that both service mesh and serverless will attract the general attention of the research community in the near future, as well as that of security research.

### Insights

**Comprehensive technological references**: the progressive adoption of new technologies in the world of microservices (such as blockchains, service meshes, and serverless) calls for dedicated investigations and reports on their impact on the security of these systems.

### *Correlation between research questions*

The amount of data collected in our dataset is large enough to represent a statistically-relevant sample. In this section, we leverage this to study correlations between our research

**Table 3 Correlation matrix among research questions (the values are percentages).**

| | Q2 | Q3 | Q4 | Q5 | Q6 | Q7 | Q8 | Q9 | Q10 | Q11 | Q12 | Q13 | Q14 | Q15 | Q16 | Q17 | Q18 | Q19 | Q20 |
|---|---|---|---|---|---|---|---|---|---|---|---|---|---|---|---|---|---|---|---|
| Q2 | | 27.11 | 32.80 | 8.10 | 13.75 | 3.19 | −7.74 | 12.36 | 0.41 | 24.68 | −4.12 | −22.74 | −8.06 | 6.27 | −3.88 | −6.71 | 8.45 | 0.19 | 0.41 |
| Q3 | 27.11 | | 28.59 | 7.37 | 18.93 | 29.11 | 7.49 | 8.68 | 12.81 | 16.69 | 8.54 | 0.75 | 5.51 | 15.10 | −6.93 | −10.82 | 3.57 | 2.26 | 12.81 |
| Q4 | 32.80 | 28.59 | | 12.18 | 6.30 | 5.05 | 10.50 | 6.05 | 8.45 | 17.28 | 9.01 | −10.39 | −7.34 | −0.42 | 1.05 | 1.34 | 6.88 | −6.90 | 8.45 |
| Q5 | 8.10 | 7.37 | 12.18 | | 6.15 | 8.26 | 12.31 | 5.58 | 13.51 | −12.44 | 5.04 | −2.41 | 5.31 | 9.24 | −10.48 | −7.12 | −13.61 | −9.07 | 8.06 |
| Q6 | 13.75 | 18.93 | 6.30 | 6.15 | | 77.49 | 22.89 | 14.23 | 14.86 | 3.83 | 5.58 | 0.88 | 17.76 | 14.23 | −0.42 | 2.99 | 6.90 | 4.96 | −5.83 |
| Q7 | 3.19 | 29.11 | 5.05 | 8.26 | 77.49 | | 20.77 | 12.55 | 17.68 | 1.97 | 0.88 | 0.78 | 15.67 | 12.55 | 4.41 | −1.48 | 10.17 | 7.44 | −5.14 |
| Q8 | −7.74 | 7.49 | 10.50 | 12.31 | 22.89 | 20.77 | | 10.03 | 14.15 | −5.27 | 20.31 | 20.12 | 31.15 | 13.76 | 8.28 | 9.01 | 4.83 | −4.89 | −8.03 |
| Q9 | 12.36 | 8.68 | 6.05 | 5.58 | 14.23 | 12.55 | 10.03 | | 25.72 | 3.19 | 13.09 | −0.84 | 15.70 | 14.01 | −0.66 | 3.25 | 8.28 | 14.01 | −3.80 |
| Q10 | 0.41 | 12.81 | 8.45 | 13.51 | 14.86 | 17.68 | 14.15 | 25.72 | | 8.88 | 10.14 | 0.37 | 7.54 | 6.04 | 5.95 | 3.53 | 2.93 | 6.04 | 12.17 |
| Q11 | 24.68 | 16.69 | 17.28 | −12.44 | 3.83 | 1.97 | −5.27 | 3.19 | 8.88 | | 0.36 | 5.13 | −17.71 | 0.15 | 12.62 | 9.16 | 9.02 | 6.24 | 8.88 |
| Q12 | −4.12 | 8.54 | 9.01 | 5.04 | 5.58 | 0.88 | 20.31 | 13.09 | 10.14 | 0.36 | | −1.44 | 9.26 | 4.38 | −1.62 | 6.16 | 1.34 | −4.32 | −2.81 |
| Q13 | −22.74 | 0.75 | −10.39 | −2.41 | 0.88 | 0.78 | 20.12 | −0.84 | 0.37 | 5.13 | −1.44 | | 26.24 | 9.08 | 11.22 | 13.12 | 7.16 | 5.77 | 5.29 |
| Q14 | −8.06 | 5.51 | −7.34 | 5.31 | 17.76 | 15.67 | 31.15 | 15.70 | 7.54 | −17.71 | 9.26 | 26.24 | | 22.90 | 10.67 | 12.47 | 11.75 | 8.50 | −3.18 |
| Q15 | 6.27 | 15.10 | −0.42 | 9.24 | 14.23 | 12.55 | 13.76 | 14.01 | 6.04 | 0.15 | 4.38 | 9.08 | 22.90 | | 13.07 | 10.85 | 18.85 | 14.01 | −3.80 |
| Q16 | −3.88 | −6.93 | 1.05 | −10.48 | −0.42 | 4.41 | 8.28 | −0.66 | 5.95 | 12.62 | −1.62 | 11.22 | 10.67 | 13.07 | | 57.34 | 25.21 | 19.94 | 0.85 |
| Q17 | −6.71 | −10.82 | 1.34 | −7.12 | 2.99 | −1.48 | 9.01 | 3.25 | 3.53 | 9.16 | 6.16 | 13.12 | 12.47 | 10.85 | 57.34 | | 33.08 | 10.85 | −2.13 |
| Q18 | 8.45 | 3.57 | 6.88 | −13.61 | 6.90 | 10.17 | 4.83 | 8.28 | 2.93 | 9.02 | 1.34 | 7.16 | 11.75 | 18.85 | 25.21 | 33.08 | | 40.00 | −4.94 |
| Q19 | 0.19 | 2.26 | −6.90 | −9.07 | 4.96 | 7.44 | −4.89 | 14.01 | 6.04 | 6.24 | −4.32 | 5.77 | 8.50 | 14.01 | 19.94 | 10.85 | 40.00 | | −3.80 |
| Q20 | 0.41 | 12.81 | 8.45 | 8.06 | −5.83 | −5.14 | −8.03 | −3.80 | 12.17 | 8.88 | −2.81 | 5.29 | −3.18 | −3.80 | 0.85 | −2.13 | −4.94 | −3.80 | |

questions, by way of the answers that the publications in our dataset give to each of them. Correlations can be used to understand which of the different aspects of microservice security are most commonly in a positive correlation (paired) in the dataset, and which ones are negatively correlated (mutually exclusive).

We report in Table 3 the correlation matrix—excluding research question Q1, since no publication answered it. While the obtained matrix is symmetric and we could report just one half, in Table 3 we report the full matrix for convenience, to provide a more immediate view of how each question correlates with all of the other ones.

We conditionally colour the cells of the Table, first, attributing colour intensity according to correlation absolute value—maximal intensity for 100% and degrading towards 0%—, second, setting a transition threshold above 30% (absolute value) from green to orange, to help to spot relevant correlations. Looking at the Table, we notice the predominance of light-coloured cells. This result can be interpreted as an indication that the research questions used in this work are mostly orthogonal, and thus suited to cover the reviewed subject with almost no wasteful overlap.

No anti-correlation was found, *i.e.*, negative correlations over the 30% threshold in absolute value. In the following, we comment on all positive correlations above 30%.

No anti-correlation was found, *i.e.*, negative correlations over the 30% threshold in absolute value. In the following, we comment on all positive correlations above 30%.

**Q2–Q4 (32.80%)** The questions relate the use of STRIDE threat model with one of its identified specific threats. This seems to be an obvious correlation since we are looking for a specific STRIDE path, or at least one of his threats.

**Q7–Q6 (77.49%)** The questions ask if the publication mentions IPS or IDS functionalities respectively. The strong correlation indicates how IPS and IDS are strictly related. Indeed, in practice, IDS may exist without IPS, but not the opposite, because prevention mechanisms are typically built as a reaction to a detected attack;

**Q8–Q14 (31.15%)** The questions relate Threat Intelligence functionalities with Infrastructure as a Service deployment, which can define a campaign strategy for a Threat Intelligence analysis.

**Q17–Q16 (57.34%)** The questions relate the Agile development methodology with DevOps and Continuous Integration. As also emphasised in other studies like *Lwakatare et al. (2019)*, this correlation can be easily explained by the fact that DevOps is sometimes considered an Agile method or its evolution. Processes adopting DevOps, therefore, adopt also Agile;

**Q19–Q18 (40.00%)** The questions relate Domain-Driven Development and Model-Driven Development. We conjecture that this correlation is present because mentions of Domain-Driven Development often mentions Model-Driven Development as an alternative approach and *vice versa*;

**Q18–Q17 (33.08%)** The questions relate Domain-Driven Development and Agile methodologies, indicating a correlation, mainly because often Agile methodologies employ Domain-Driven Development.

## Threats to validity

Our study is subject to limitations that can be categorised into construct validity, external validity, internal validity, and reliability following the guidelines of *Runeson et al. (2012)*.

*Construct validity* "reflects to what extent the operational measures that are studied really represent what the researcher has in mind and what is investigated according to the research questions". To mitigate a potential misinterpretation and making sure that the constructs discussed in the research questions are not interpreted differently by the researchers, we adopted various triangulation rounds using online meetings and we designed a set of binary research questions to foster objectivity in answering them.

Another potential risk regards whether we were exhaustive during data collection, *i.e.*, whether we may have missed any significant publication in our review. This risk cannot be completely mitigated but to minimise this risk we deliberately chose to have simple and broad keywords giving more initial hits that later were further filtered out. Moreover, we conducted a snowballing process to extend our initial dataset looking for potentially relevant publications that our query did not select.

*External validity* regards the applicability of a set of results in a more general context and is not a concern for this study since we focus on the intersection of the fields of microservices and security without any attempt of generalising the findings to a broader context. We do not claim that either our qualitative or our quantitative findings should also hold for other large fields.

*Internal validity* is of concern when causal relations are examined when there is a risk that the investigated factor is also affected by a third factor. This thread is not a concern for this study because we presented only correlations between different factors but did not examine causal relations.

*Reliability* concerns to what extent the data collection and analysis depend on the actual researchers. This risk has been partially mitigated by selecting as many objective criteria as possible for the filtering and by requiring at least a two-people consensus in case of more subjective decisions. In particular, the retrieval of the publications was performed by using search engines. The first filtering of the results (Step 2, cf. "Review Method") was conducted by running a script that uses objective criteria such as counting the number of keywords present and the length of the publication. These automatically computed results were double-checked by at least one author to prevent problems due to the parsing of PDFs and to make sure that the language of the publication was English. The second filtering (Step 3, cf. "Review Method") performed by reading the title, abstract, and (if needed) the body of the publication, was performed in parallel by two authors. Decision conflicts were solved by discussion involving at least two authors until a consensus was reached. For the publication analysis (Step 4, cf. "Review Method"), due to the binary nature and formulation of the questions, the 20 research questions were answered by the author assigned to the publication. To detect possible observer bias and errors, we selected a random subset of 15 papers and had a different author answer to the research questions. The calculation of the kappa index of agreement as proposed in *Cohen (1960)* over the two result sets yielded a value of $\kappa = 0.99998$, giving us statistical confidence over the perceived precision of questions and objectiveness of answers.

The reliability of the study is strengthened by being open and explicit about the process of data collection and analysis. For transparency, reproducibility, and reuse, we report the data used in this study at https://doi.org/10.5281/zenodo.4774894, which includes both the final dataset with the answers to all the research questions and also the set of rejected publications along with the reason for exclusion.

We also report in the Appendix each entry of our dataset and its answers to our research questions.

## DISCUSSION AND FUTURE DIRECTIONS

In this article, we presented a systematic review of the literature regarding microservice security. To conduct our research, we followed a structured approach that allowed us to gather 290 peer-reviewed publications, which, at the time of writing, constitutes the largest curated dataset on the topic.

To study our dataset, we conducted first an investigation on the metadata of the publications, which gave us some insight to map what are the publication outlets, the communities, and the key research concepts that characterise the field. Then, we performed an analysis, associating each element in our dataset to a vector of 20 different markers—presented in the form of 20 research questions.

Since our markers belong in four micro-groups (of threat-model, security, infrastructure, and development approaches), we used that partition to provide an

overview of the literature through the lenses of each cluster. As a byproduct of our analysis on the content of each publication, we found concepts and topics that we did not include in our questions but that recur in multiple publications, *e.g.*, the usage of blockchain or service-mesh technologies. To provide a more comprehensive picture of the field, we described and contextualised also these additional elements. Since our dataset forms a statistically relevant vector field, we also performed a correlation study over the components of the vectors and reported the strongest correlations (*e.g.*, between intrusion-detection (IDS) and intrusion-prevention (IPS) systems in microservice deployments) along with possible explanations of the identified phenomena.

In the following, we draw a summary of the main open challenges that emerged from our study, which forms a call for action for the community of researchers and practitioners working in the field of microservice security and its neighbouring areas.

**Data provenance:** the distributed nature of microservices calls for the certification of their outputs, which other federated services receive as input and need to trust. However, there is a lack of best practices and/or standards for such a task.

**Technology transfer:** there exists a sensible amount of research on microservices security, but transferring those results—*e.g.*, viable methods and tools for validation and verification—to the industry is difficult and applications are almost non-existent.

**Security-by-design adoption:** while many advocate for adopting security-by-design at all stages of a microservice lifecycle (from design to monitoring), there are no established references nor guidelines on how these principles can be reliably adopted in practice.

**Dedicated attack trees and threat models:** threats in microservice systems can come from multiple sources, from the interaction of the layers of a chosen technology stack to how microservices interact with each other—*e.g.*, in an exclusive network, on a federated basis, on the Web, *etc.* Practitioners lack dedicated attack trees and threat models to help them consider and tackle the multifaceted attack surface of microservice architectures.

**Comprehensive technological references:** microservice development entails the use of (heterogeneous) technology stacks, whose combinations and interactions give way to exploits at different levels. These include data leakage due to host-container interactions, threats to encryption reliability due to interacting heterogeneous standards and data-format conversions, as well as surreptitious attacks through software libraries hijacking. Besides the lack of dedicated threat models, there is also a need for concrete references to secure specific technology stacks.

**Migration to microservices:** several works provide structures and methods to migrate legacy systems to microservices architectures. However, there are no established techniques to elicit the assumptions and invariants (*e.g.*, on shared-memory communication, runtime environment, concurrent/interleaved database accesses, *etc.*) of the legacy system that the developers of the microservices must deal with—least of all considering how those factors impact the security aspects of the migrated architecture. An additional step in this direction would benefit from following principled security-by-design disciplines.

**Global view/control:** the distributed nature of microservices makes it difficult to check the correct implementation of architecture-wide security policies, especially when each

microservice has a dedicated security configuration. The issue is further exacerbated by the DevOps practice of having different teams deal separately with all aspects of the microservices they develop, including the implementations of their security policies. This fact highlights the need for tools that provide global overviews and guarantees on the security policies, protocols, and invariants of microservice systems.

**React & recover techniques:** while the literature on preventive and detective measures against attacks abound, little has been done on how microservices should react to attacks and, as a consequence, recover their normal behaviour.

**DevSecOps:** Agile and DevOps practices are widely used when developing microservices, yet there is no established reference on how these approaches should integrate security in all their aspects (from team culture, management and communication to develop technologies and techniques) and into the lifecycle of microservices.

**Fragmentation of outlets:** researchers (and practitioners) working on microservices security do not have reference venues (neither journals nor conferences). This has at least two negative consequences. First, it makes it more difficult to gather the relevant work that constitutes the current state-of-the-art of their field—a need to which this study provides a partial solution, in the form of a snapshot of the current field landscape. Second, reference venues work also as gathering and exchange points for researchers to discuss current problems and new ideas, form interest groups, and concretise new contributions and projects to advance the knowledge in the field. Here, our call for action is at the community level, advocating for the establishment of a few reference, high-quality venues able to focus, inform, and orient the agenda of the field.

Regarding the future steps of the line of work of this contribution, we notice that here we focused our investigation on peer-reviewed publications. However, in the general field of microservices (and their security, by extension) the grey literature—which includes non-peer-reviewed reports, working papers, government documents (*e.g.*, those by NIST), white papers—constitutes a relevant body of knowledge that deserves separate studies. As future work, we intend to pursue an activity similar to what we presented in this work, but purposed to investigate the grey literature.

## APPENDIX

Dataset and Research Questions in tabular form

We partition the dataset into four tables, each representing the categorisation described in "Types of Publications"— (i) Theoretical, (ii) Applicative, and (iii) Theoretical and Applicative publications and iv) Survey. For each table we have 5 columns. The first 4 columns from the left (after the column containing the reference ("Ref.") to the publication from the publications dataset) and grouped under the column group "Group" report the 4 Research Questions Groups as defined in "Research Questions". The value shown indicates the amount of questions of each group the publications answered. The last column labeled "Q.Num." presents the number of questions having a positive answer.

| Ref. | Group | | | | Q. Num. |
|------|-------|---|---|---|---------|
| | **G1** | **G2** | **G3** | **G4** | |
| Survey Publications | | | | | |
| *Sultan, Ahmad & Dimitriou (2019)* | 3 | 2 | 2 | 1 | 2,4,5,8,11,13,14,16 |
| *Cerny & Donahoo (2016)* | 0 | 0 | 1 | 0 | 13 |
| *Westerlund & Kratzke (2018)* | 0 | 1 | 1 | 1 | 11,15,16 |
| *Bandeira et al. (2019)* | 0 | 0 | 2 | 0 | 13,14 |
| *Ahmed et al. (2019)* | 0 | 1 | 1 | 0 | 8,13 |
| *Di Salle, Gallo & Pompilio (2016)* | 0 | 0 | 1 | 1 | 13,16 |
| *Bélair, Laniepce & Menaud (2019)* | 0 | 0 | 2 | 0 | 13,14 |
| *Márquez & Astudillo (2019)* | 2 | 0 | 2 | 0 | 2,4,13,14 |
| *Puliafito et al. (2019)* | 1 | 0 | 0 | 0 | 2 |
| *Manu et al. (2016)* | 0 | 2 | 1 | 1 | 8,11,13,17 |
| *Lysne et al. (2016)* | 2 | 0 | 1 | 0 | 3,4,13 |
| *Panduman, Sukaridhoto & Tjahjono (2019)* | 1 | 0 | 0 | 0 | 2 |
| *Casale et al. (2016)* | 0 | 0 | 1 | 1 | 13,16 |
| *Soldani, Tamburri & Van Den Heuvel (2018)* | 1 | 1 | 2 | 3 | 4,8,13,14,16–18 |
| *Almeida et al. (2017)* | 1 | 0 | 1 | 0 | 2,13 |
| *Yousefpour et al. (2019)* | 0 | 1 | 1 | 0 | 11,13 |
| *Trnka, Černý & Stickney (2018)* | 0 | 0 | 1 | 0 | 13 |
| *Adedugbe et al. (2019)* | 1 | 1 | 2 | 0 | 3,8,13,14 |
| *Lichtenthäler et al. (2019)* | 0 | 1 | 1 | 0 | 11,13 |
| *Mohsin & Janjua (2018)* | 0 | 1 | 1 | 1 | 11,13,17 |
| *Noura, Atiquzzaman & Gaedke (2019)* | 0 | 0 | 1 | 0 | 13 |
| *Rao et al. (2018)* | 0 | 1 | 1 | 0 | 11,13 |
| *Yang et al. (2014)* | 1 | 1 | 1 | 0 | 3,11,13 |
| *Yu et al. (2019)* | 1 | 1 | 2 | 1 | 2,8,13,14,16 |
| *Casalicchio & Iannucci (2020)* | 2 | 1 | 1 | 0 | 2,5,11,13 |
| *Plaza, Daz & Pérez (2018)* | 0 | 0 | 0 | 1 | 17 |
| *Di Francesco, Malavolta & Lago (2017)* | 2 | 0 | 0 | 2 | 4,5,16,17 |
| *Islam, Manivannan & Zeadally (2016)* | 3 | 1 | 0 | 0 | 2,4,5,8 |
| *Vale et al. (2019)* | 2 | 2 | 0 | 0 | 2,5,9,11 |
| *Bélair, Laniepce & Menaud (2019)* | 0 | 0 | 2 | 0 | 13,14 |
| *Márquez & Astudillo (2019)* | 2 | 0 | 2 | 0 | 2,4,13,14 |
| *Puliafito et al. (2019)* | 1 | 0 | 0 | 0 | 2 |
| *Manu et al. (2016)* | 0 | 2 | 1 | 1 | 8,11,13,17 |
| *Lysne et al. (2016)* | 2 | 0 | 1 | 0 | 3,4,13 |
| *Panduman, Sukaridhoto & Tjahjono (2019)* | 1 | 0 | 0 | 0 | 2 |
| *Casale et al. (2016)* | 0 | 0 | 1 | 1 | 13,16 |
| *Soldani, Tamburri & Van Den Heuvel (2018)* | 1 | 1 | 2 | 3 | 4,8,13,14,16,17,18 |
| *Almeida et al. (2017)* | 1 | 0 | 1 | 0 | 2,13 |
| *Yousefpour et al. (2019)* | 0 | 1 | 1 | 0 | 11,13 |
| *Sultan, Ahmad & Dimitriou (2019)* | 3 | 2 | 2 | 1 | 2,4,5,8,11,13,14,16 |

| (continued) | | | | | |
|---|---|---|---|---|---|
| **Ref.** | **Group** | | | | **Q. Num.** |
| | **G1** | **G2** | **G3** | **G4** | |
| *Ahmed et al. (2019)* | 0 | 1 | 1 | 0 | 8,13 |
| *Trnka, Černý & Stickney (2018)* | 0 | 0 | 1 | 0 | 13 |
| *Cerny & Donahoo (2016)* | 0 | 0 | 1 | 0 | 13 |
| *Ahmadvand et al. (2018)* | 2 | 7 | 3 | 3 | 2,3,6–18 |
| *Adedugbe et al. (2019)* | 1 | 1 | 2 | 0 | 3,8,13,14 |
| *Lichtenthäler et al. (2019)* | 0 | 1 | 1 | 0 | 11,13 |
| *Mohsin & Janjua (2018)* | 0 | 1 | 1 | 1 | 11,13,17 |
| *Niazi, Mishra & Gill (2018)* | 0 | 0 | 0 | 0 | |
| *Noura, Atiquzzaman & Gaedke (2019)* | 0 | 0 | 1 | 0 | 13 |
| *Rao et al. (2018)* | 0 | 1 | 1 | 0 | 11,13 |
| *Yang et al. (2014)* | 1 | 1 | 1 | 0 | 3,11,13 |
| *Yu et al. (2019)* | 1 | 1 | 2 | 1 | 2,8,13,14,16 |
| *Casalicchio & Iannucci (2020)* | 2 | 1 | 1 | 0 | 2,5,11,13 |
| *Plaza, Daz & Pérez (2018)* | 0 | 0 | 0 | 1 | 17 |
| *Di Francesco, Malavolta & Lago (2017)* | 2 | 0 | 0 | 2 | 4,5,16,17 |
| *Westerlund & Kratzke (2018)* | 0 | 1 | 1 | 1 | 11,15,16 |
| *Islam, Manivannan & Zeadally (2016)* | 3 | 1 | 0 | 0 | 2,4,5,8 |
| *Vale et al. (2019)* | 2 | 2 | 0 | 0 | 2,5,9,11 |
| *Lie, Sánchez-Gordón & Colomo-Palacios (2020)* | 2 | 1 | 0 | 3 | 3,4,11,16,17,20 |
| *Ali, Caprolu & Pietro (2020)* | 4 | 0 | 0 | 0 | 2,3,4,5 |
| *de Sousa et al. (2020)* | 2 | 1 | 1 | 2 | 2,3,11,13,16,19 |
| *Adam & Alam (2020)* | 2 | 0 | 0 | 0 | 2,5 |
| *Delicato et al. (2020)* | 3 | 0 | 0 | 0 | 2,4,5 |
| *Mohamed, Challenger & Kardas (2020)* | 2 | 0 | 0 | 1 | 2,4,18 |
| *Waseem, Liang & Shahin (2020)* | 2 | 1 | 2 | 4 | 2,4,11,13,14,16–19 |
| *Mishra & Otaiwi (2020)* | 1 | 1 | 1 | 2 | 2,11,13,16,17 |
| *Niknejad et al. (2020)* | 1 | 1 | 0 | 0 | 2,11 |
| *Moura & Hutchison (2020)* | 2 | 0 | 0 | 0 | 2,4 |
| *de Araujo Zanella, da Silva & Albini (2020)* | 4 | 2 | 0 | 0 | 2,3,4,5,6,7 |
| *Mohamed, Challenger & Kardas (2020)* | 2 | 0 | 0 | 1 | 2,4,18 |
| *Razzaq (2020)* | 2 | 1 | 3 | 4 | 2,3,11,13–19 |
| *Wu et al. (2019)* | 1 | 2 | 0 | 0 | 2,6,11 |
| **Ref.** | **Group** | | | | **Q. Num.** |
| | **G1** | **G2** | **G3** | **G4** | |
| Applicative Publications | | | | | |
| *George & Mahmoud (2017)* | 2 | 0 | 1 | 0 | 2,5,13 |
| *Thramboulidis, Vachtsevanou & Kontou (2019)* | 1 | 1 | 1 | 0 | 5,8,13 |
| *Ciavotta et al. (2017)* | 0 | 1 | 1 | 1 | 11,13,17 |
| *Morris (2017)* | 2 | 2 | 1 | 0 | 3,5,8,11,13 |
| *Fetzer et al. (2017)* | 2 | 3 | 2 | 1 | 2,3,6–8,13,14,16 |

(Continued)

| (continued) | | | | | |
|---|---|---|---|---|---|
| **Ref.** | **Group** | | | | **Q. Num.** |
| | **G1** | **G2** | **G3** | **G4** | |
| *Jita & Pieterse (2018)* | 1 | 0 | 2 | 0 | 2,13,14 |
| *Perrone & Romano (2017)* | 0 | 0 | 1 | 1 | 13,17 |
| *Pahl & Aubet (2018)* | 1 | 0 | 2 | 0 | 3,13,14 |
| *Sialm & Knittl (2016)* | 0 | 0 | 1 | 0 | 13 |
| *Du, Xie & He (2018)* | 1 | 1 | 1 | 0 | 3,8,13 |
| *Kalske, Mäkitalo & Mikkonen (2017)* | 1 | 1 | 2 | 1 | 2,8,13,14,16 |
| *Nehme et al. (2018)* | 1 | 1 | 1 | 0 | 2,8,13 |
| *Nikoloudakis et al. (2019)* | 0 | 2 | 1 | 0 | 8,11,13 |
| *Salomoni et al. (2018)* | 0 | 0 | 2 | 1 | 13,14,16 |
| *Stallenberg & Panichella (2019)* | 3 | 1 | 0 | 0 | 2,3,5,7 |
| *Morris (2017)* | 2 | 2 | 1 | 0 | 3,5,8,11,13 |
| *Fetzer et al. (2017)* | 2 | 3 | 2 | 1 | 2,3,6,7,8,13,14,16 |
| *Jita & Pieterse (2018)* | 1 | 0 | 2 | 0 | 2,13,14 |
| *Perrone & Romano (2017)* | 0 | 0 | 1 | 1 | 13,17 |
| *Pahl & Aubet (2018)* | 1 | 0 | 2 | 0 | 3,13,14 |
| *Sialm & Knittl (2016)* | 0 | 0 | 1 | 0 | 13 |
| *Cerny & Donahoo (2016)* | 0 | 0 | 1 | 0 | 13 |
| *Du, Xie & He (2018)* | 1 | 1 | 1 | 0 | 3,8,13 |
| *Kalske, Mäkitalo & Mikkonen (2017)* | 1 | 1 | 2 | 1 | 2,8,13,14,16 |
| *Nehme et al. (2018)* | 1 | 1 | 1 | 0 | 2,8,13 |
| *Nikoloudakis et al. (2019)* | 0 | 2 | 1 | 0 | 8,11,13 |
| *Salomoni et al. (2018)* | 0 | 0 | 2 | 1 | 13,14,16 |
| *Stallenberg & Panichella (2019)* | 3 | 1 | 0 | 0 | 2,3,5,7 |
| *Park & Jeon (2020)* | 1 | 1 | 1 | 0 | 2,11,13 |
| *Xu & Bian (2020)* | 0 | 1 | 1 | 1 | 11,13,16 |
| *Brondolin & Santambrogio (2020)* | 1 | 2 | 1 | 0 | 2,6,11,13 |
| *Ma et al. (2020)* | 1 | 1 | 0 | 1 | 3,7,16 |
| *Olsthoorn, van Deursen & Panichella (2020)* | 1 | 1 | 0 | 1 | 3,7,16 |
| *Chen, Chen & Yu (2020)* | 2 | 1 | 0 | 0 | 2,3,11 |
| *Zuo et al. (2020)* | 2 | 1 | 1 | 0 | 2,3,11,13 |
| *Luntovskyy & Shubyn (2020)* | 1 | 3 | 1 | 2 | 2,6,7,11,13,16,19 |
| *Ghuge et al. (2020)* | 3 | 1 | 0 | 0 | 2,3,4,11 |
| *Bobel, Gerostathopoulos & Bures (2020)* | 1 | 1 | 0 | 0 | 2,11 |
| *Zhang et al. (2020)* | 1 | 1 | 0 | 0 | 2,11 |
| *Hang, Ullah & Kim (2020)* | 3 | 1 | 0 | 0 | 2,3,4,11 |
| *Forti, Ferrari & Brogi (2020)* | 3 | 3 | 1 | 0 | 2,3,4,6,7,11,13 |
| *Stock, Schel & Bauernhansl (2020)* | 2 | 0 | 0 | 1 | 2,4,18 |
| *Hasan & Starly (2020)* | 1 | 0 | 1 | 0 | 2,13 |
| *Kallergis et al. (2020)* | 2 | 1 | 1 | 0 | 2,4,11,13 |
| *Amir-Mohammadian & Kari (2020)* | 2 | 1 | 0 | 0 | 2,4,11 |

| (continued) | | | | | |
|---|---|---|---|---|---|
| **Ref.** | **Group** | | | | **Q. Num.** |
| | **G1** | **G2** | **G3** | **G4** | |
| *Roca et al. (2020)* | 3 | 1 | 1 | 0 | 2,3,4,11,13 |
| *Bromberg & Gitzinger (2020)* | 2 | 4 | 1 | 2 | 2,3,6–8,11,13,16,18 |
| *Jaworski, Karwowski & Rusek (2019)* | 2 | 1 | 1 | 0 | 2,4,11,13 |
| **Ref.** | **Group** | | | | **Q. Num.** |
| | **G1** | **G2** | **G3** | **G4** | |
| Theoretical Publications (1/3) | | | | | |
| *ShuLin & JiePing (2020)* | 3 | 1 | 1 | 0 | 2,3,4,11,13 |
| *Dilshan et al. (2020)* | 4 | 1 | 1 | 0 | 2,3,4,5,11,13 |
| *Flora (2020)* | 3 | 3 | 1 | 0 | 2,3,4,6,7,11,13 |
| *Flora, Gonçalves & Antunes (2020)* | 3 | 3 | 0 | 0 | 2,3,4,6,7,11 |
| *Bogatinovski et al. (2020)* | 2 | 0 | 0 | 0 | 2,4 |
| *Damis et al. (2020)* | 3 | 1 | 0 | 0 | 2,3,4,11 |
| *Iraqi & El Bakkali (2020)* | 3 | 1 | 1 | 0 | 2,3,4,11,13 |
| *Dewanta (2020)* | 3 | 1 | 1 | 0 | 2,3,4,11,13 |
| *Bumblauskas et al. (2020)* | 1 | 0 | 0 | 0 | 2 |
| *Giaimo, Andrade & Berger (2020)* | 2 | 0 | 0 | 2 | 2,4,16,17 |
| *Lenarduzzi et al. (2020)* | 2 | 1 | 1 | 2 | 2,4,11,13,16,17 |
| *Mann (2020)* | 3 | 1 | 0 | 1 | 2,3,4,9,16 |
| *Costa, Pires & Delicato (2020)* | 0 | 1 | 1 | 3 | 11,13,16,17,18 |
| *Fahmideh & Zowghi (2020)* | 1 | 0 | 0 | 2 | 2,18,19 |
| *Razian, Fathian & Buyya (2020)* | 1 | 0 | 0 | 0 | 2 |
| *Taherizadeh & Grobelnik (2020)* | 0 | 1 | 1 | 0 | 11,13 |
| *Safaryan et al. (2020)* | 2 | 1 | 0 | 0 | 2,4,11 |
| *de Toledo, Martini & Sjøberg (2020)* | 0 | 1 | 1 | 1 | 11,13,16 |
| *Alulema et al. (2020)* | 0 | 1 | 1 | 3 | 11,13,16,17,19 |
| *Kapferer & Zimmermann (2020)* | 0 | 0 | 1 | 2 | 13,16,19 |
| *Redelinghuys, Basson & Kruger (2019)* | 2 | 1 | 0 | 0 | 2,3,11 |
| *Dash et al. (2020)* | 2 | 1 | 0 | 0 | 2,3,11 |
| *Kwon et al. (2020)* | 3 | 1 | 0 | 0 | 2,3,4,11 |
| *Khan & Shameem (2020)* | 1 | 1 | 1 | 3 | 2,11,13,16,17,18 |
| *DesLauriers et al. (2020)* | 1 | 1 | 1 | 2 | 2,11,13,16,17 |
| *Bertolino et al. (2020)* | 0 | 1 | 1 | 1 | 11,13,16 |
| *Di Sanzo, Avresky & Pellegrini (2021)* | 2 | 1 | 0 | 1 | 2,4,11,16 |
| *Moreira et al. (2020)* | 2 | 1 | 1 | 2 | 2,4,11,13,16,17 |
| *Li et al. (2019b)* | 2 | 1 | 1 | 0 | 2,3,11,13 |
| **Ref.** | **Group** | | | | **Q. Num.** |
| | **G1** | **G2** | **G3** | **G4** | |
| Theoretical Publications (2/3) | | | | | |
| *Callegati et al. (2016)* | 2 | 4 | 1 | 0 | 3,4,6,7,8,11,13 |

(Continued)

| (continued) | | | | | |
|---|---|---|---|---|---|
| Ref. | Group | | | | Q. Num. |
| | G1 | G2 | G3 | G4 | |
| *Preuveneers & Joosen (2019)* | 1 | 0 | 2 | 1 | 5,13,14,16 |
| *Abidi et al. (2019)* | 3 | 1 | 2 | 0 | 3,4,5,8,13,14 |
| *Baboi, Iftene & Gfu (2019)* | 0 | 0 | 1 | 0 | 13 |
| *He & Yang (2017)* | 1 | 1 | 0 | 1 | 3,11,16 |
| *Sim, Barus & Jaya (2019)* | 0 | 1 | 0 | 0 | 11 |
| *Brito et al. (2019)* | 1 | 3 | 1 | 0 | 4,8,11,12,13 |
| *Niazi, Mishra & Gill (2018)* | 0 | 0 | 0 | 1 | 17 |
| *Lu et al. (2017)* | 3 | 0 | 2 | 0 | 2,3,5,13,15 |
| *Beekman & Porter (2017)* | 2 | 0 | 1 | 0 | 2,5,13 |
| *Syed & Fernandez (2017)* | 3 | 0 | 2 | 0 | 2,4,5,13,15 |
| *Syed & Fernandez (2018)* | 2 | 0 | 3 | 1 | 2,4,13–16 |
| *Bhattacharya (2019)* | 0 | 3 | 1 | 0 | 6,7,8,13 |
| *Zhang et al. (2017)* | 0 | 2 | 3 | 0 | 6,7,13–15 |
| *Zaheer et al. (2019)* | 1 | 0 | 1 | 0 | 5,13 |
| *Walsh & Manferdelli (2017)* | 0 | 0 | 1 | 0 | 13 |
| *Torkura, Sukmana & Meinel (2017)* | 1 | 2 | 2 | 2 | 2,6,11,13,14,16,17 |
| *Clancy, McGwier & Chen (2019)* | 0 | 0 | 2 | 0 | 13,14 |
| *Cerny, Sedlisky & Donahoo (2018)* | 0 | 0 | 2 | 2 | 13,14,18,19 |
| *Tourani et al. (2019)* | 1 | 3 | 3 | 0 | 3,6,7,8,13–15 |
| *Chen, Huang & Chen (2019)* | 1 | 1 | 1 | 0 | 5,8,13 |
| *Anisetti et al. (2019)* | 0 | 0 | 1 | 2 | 15,16,17 |
| *Leite et al. (2019)* | 0 | 0 | 2 | 2 | 13,14,16,17 |
| *Suneja, Kanso & Isci (2019)* | 0 | 2 | 1 | 0 | 8,11,13 |
| *Schlossnagle (2018)* | 0 | 0 | 1 | 2 | 13,16,17 |
| *Schlossnagle (2017)* | 0 | 0 | 1 | 2 | 13,16,17 |
| *Guija & Siddiqui (2018)* | 4 | 0 | 1 | 0 | 2,3,4,5,13 |
| *Esparrachiari, Reilly & Rentz (2018)* | 2 | 0 | 0 | 0 | 2,5 |
| *Gupta, Venkatachalapathy & Jeberla (2019)* | 0 | 0 | 0 | 2 | 16,17 |
| *Troiano et al. (2019)* | 2 | 0 | 2 | 0 | 2,3,13,14 |
| *Tchoubraev & Wiczynski (2015)* | 0 | 1 | 0 | 0 | 11 |
| *Sun, Nanda & Jaeger (2015)* | 2 | 5 | 2 | 0 | 3,5–8,10,11,13,14 |
| *Thanh et al. (2016)* | 2 | 3 | 1 | 2 | 2,5,6,7,8,13,16,17 |
| *Ahmadvand & Ibrahim (2016)* | 0 | 0 | 1 | 0 | 13 |
| *Kelbert et al. (2017)* | 0 | 0 | 2 | 0 | 13,14 |
| *Esposito et al. (2017)* | 1 | 0 | 1 | 0 | 2,13 |
| *Torkura et al. (2017)* | 2 | 3 | 1 | 2 | 4,5,8,11,12,14,16,17 |
| *Yarygina & Bagge (2018)* | 1 | 0 | 1 | 2 | 4,13,17,18 |
| *Trihinas et al. (2018)* | 0 | 0 | 1 | 1 | 13,16 |
| *Bánáti et al. (2018)* | 3 | 0 | 2 | 0 | 2,3,5,13,14 |
| *Pahl, Aubet & Liebald (2018)* | 1 | 0 | 1 | 0 | 5,13 |

| Ref. | Group | | | | Q. Num. |
|------|-------|------|------|------|---------|
| | G1 | G2 | G3 | G4 | |
| *Diekmann et al. (2018)* | 0 | 0 | 1 | 1 | 13,17 |
| *Trihinas, Tryfonos & Dikaiakos (2016)* | 0 | 0 | 2 | 1 | 13,14,16 |
| *Nehme et al. (2019)* | 0 | 0 | 1 | 2 | 13,16,17 |
| *Torkura, Sukmana & Kayem (2018)* | 1 | 1 | 1 | 0 | 4,8,13 |
| *Gerking & Schubert (2019)* | 1 | 0 | 1 | 0 | 5,13 |
| *Bogner et al. (2019)* | 0 | 0 | 1 | 4 | 13,16–19 |
| *Petrovska, Memeti & Imeri (2019)* | 2 | 0 | 1 | 0 | 2,5,14 |
| *Osman et al. (2019)* | 1 | 2 | 2 | 0 | 5,6,7,13,14 |
| *Chen (2019)* | 1 | 1 | 1 | 0 | 4,8,13 |
| *Wu et al. (2019)* | 1 | 1 | 1 | 0 | 2,11,14 |
| *Li et al. (2019b)* | 1 | 1 | 0 | 0 | 2,11 |
| *Mansfield-Devine (2018)* | 1 | 1 | 1 | 1 | 4,8,13,16 |
| *Trubiani et al. (2018)* | 3 | 1 | 0 | 1 | 2,4,5,8,16 |
| *Krämer, Frese & Kuijper (2019)* | 1 | 0 | 1 | 0 | 5,13 |
| *Varghese & Buyya (2018)* | 0 | 0 | 1 | 0 | 13 |
| *Elsayed & Zulkernine (2019)* | 1 | 1 | 1 | 0 | 4,8,13 |
| *Reyna et al. (2018)* | 1 | 0 | 1 | 0 | 2,13 |
| *Vaquero et al. (2019)* | 0 | 0 | 1 | 0 | 13 |
| *Kochovski et al. (2019)* | 0 | 0 | 2 | 0 | 13,14 |
| *Lwakatare et al. (2019)* | 0 | 2 | 0 | 2 | 8,11,16,17 |
| *Avritzer et al. (2020)* | 1 | 2 | 0 | 0 | 2,6,8 |
| *Nagothu et al. (2018)* | 1 | 1 | 0 | 0 | 2,11 |
| *Baker & Nguyen (2019)* | 3 | 2 | 0 | 0 | 2,4,5,9,11 |
| *Buzachis & Villari (2018)* | 1 | 1 | 0 | 0 | 2,11 |
| *Yuan et al. (2019)* | 0 | 0 | 2 | 0 | 13,14 |
| *Preuveneers & Joosen (2017)* | 1 | 0 | 1 | 0 | 5,13 |
| *Taha, Talhi & Ould-Slimanec (2019)* | 1 | 1 | 1 | 0 | 2,8,13 |
| *De Donno et al. (2019)* | 1 | 3 | 2 | 0 | 2,8,9,11,13,14 |
| *Ghayyur et al. (2018)* | 0 | 2 | 1 | 0 | 8,11,13 |
| *Xu, Jin & Kim (2019)* | 2 | 1 | 1 | 0 | 2,3,8,13 |
| *Zhiyi, Shahidehpour & Xuan (2018)* | 0 | 0 | 1 | 0 | 13 |
| *Zimmermann (2017b)* | 0 | 1 | 1 | 3 | 11,13,16–18 |
| *Tien et al. (2019)* | 2 | 1 | 0 | 0 | 2,5,6 |
| *Oppermann et al. (2018)* | 0 | 0 | 1 | 0 | 13 |
| *Brucker et al. (2017)* | 1 | 0 | 1 | 0 | 5,13 |
| *Krishnan, Duttagupta & Achuthan (2019)* | 2 | 0 | 2 | 1 | 2,5,13,14,17 |
| *Salibindla (2018)* | 1 | 0 | 0 | 0 | 2 |
| *Nguyen & Baker (2019)* | 1 | 0 | 1 | 0 | 2,13 |
| *Pustchi, Krishnan & Sandhu (2015)* | 4 | 1 | 2 | 0 | 2–6,13,15 |
| *Westerlund & Kratzke (2018)* | 0 | 1 | 1 | 1 | 11,15,16 |

(Continued)

| (continued) | | | | | |
|---|---|---|---|---|---|
| **Ref.** | **Group** | | | | **Q. Num.** |
| | **G1** | **G2** | **G3** | **G4** | |
| *Garg & Garg (2019)* | 1 | 1 | 0 | 2 | 5,11,16,17 |
| *Souppaya, Morello & Scarfone (2017)* | 3 | 3 | 1 | 2 | 2,4,5,8,10,11,13,16,17 |
| *Brenner et al. (2017)* | 0 | 1 | 0 | 0 | 11 |
| *Vassilakis, Panaousis & Mouratidis (2016)* | 0 | 0 | 0 | 0 | |
| *Yarygina (2018)* | 1 | 3 | 2 | 2 | 2,6,8,11,13,14,16,17 |
| *Bozan, Lyytinen & Rose (2020)* | 1 | 0 | 1 | 2 | 2,13,16,17 |
| *Cleveland et al. (2020)* | 1 | 0 | 0 | 0 | 2 |
| *Reed (2020)* | 1 | 0 | 0 | 0 | 2 |
| *Baarzi et al. (2020)* | 3 | 3 | 1 | 0 | 2-,6,7,11,13 |
| *Li et al. (2020)* | 1 | 1 | 1 | 2 | 2,11,13,16,17 |
| *Sundelin, Gonzalez-Huerta & Wnuk (2020)* | 1 | 1 | 0 | 0 | 2,11 |
| *Sharma, Lawrenz & Rausch (2020)* | 2 | 1 | 1 | 0 | 2,5,11,13 |
| *Walker & Cerny (2020)* | 1 | 1 | 1 | 0 | 2,11,13 |
| *Leite et al. (2020)* | 1 | 1 | 1 | 2 | 2,11,13,16,17 |
| *Russinovich et al. (2021)* | 3 | 1 | 1 | 3 | 2,3,4,11,13,16-18 |
| *Mohammed & Mohammed (2020)* | 4 | 2 | 0 | 0 | 2,3,4,5,6,7 |
| *de Oliveira Rosa et al. (2020)* | 0 | 1 | 2 | 2 | 11,13,14,16,17 |
| *Ke, Wu & Yang (2020)* | 2 | 0 | 0 | 0 | 2,4 |
| *Tuma et al. (2020)* | 3 | 1 | 0 | 0 | 2,3,4,6 |
| *Wieber (2020)* | 3 | 1 | 1 | 3 | 2-4,11,13,16-18 |
| *Hajek et al. (2020)* | 3 | 1 | 1 | 0 | 2,3,4,11,13 |
| *Chondamrongkul, Sun & Warren (2020)* | 4 | 1 | 1 | 0 | 2,3,4,5,11,13 |
| *Liang & Zhao (2020)* | 3 | 1 | 0 | 0 | 2,3,4,11 |
| *Liu et al. (2020)* | 0 | 1 | 1 | 0 | 11,13 |
| *Gorige et al. (2020)* | 3 | 1 | 1 | 0 | 2,3,4,11,13 |
| *Cerny et al. (2020)* | 1 | 1 | 1 | 1 | 2,11,13,18 |
| *Tenev & Tsvetanov (2020)* | 2 | 1 | 1 | 0 | 2,4,11,13 |
| *Jin et al. (2020)* | 2 | 1 | 1 | 0 | 2,4,11,13 |
| *Wang et al. (2020)* | 2 | 1 | 1 | 0 | 2,4,11,13 |
| **Ref.** | **Group** | | | | **Q. Num.** |
| | **G1** | **G2** | **G3** | **G4** | |
| Theoretical Publications (3/3) | | | | | |
| *Badii et al. (2019)* | 2 | 2 | 1 | 0 | 3,5,11-13 |
| *Yang et al. (2018)* | 1 | 1 | 1 | 0 | 4,8,13 |
| *Kang et al. (2018)* | 0 | 0 | 2 | 0 | 13,14 |
| *Casale et al. (2019)* | 0 | 0 | 2 | 3 | 13,14,16-18 |
| *Di Ciccio et al. (2019)* | 0 | 2 | 1 | 1 | 9,10,14,19 |
| *Kathiravelu, Van Roy & Veiga (2019)* | 0 | 1 | 1 | 0 | 11,13 |
| *Łaskawiec, Choraś & Kozik (2019)* | 0 | 1 | 1 | 0 | 11,13 |
| *Leite et al. (2017)* | 0 | 1 | 1 | 0 | 11,13 |

| (continued) | | | | | |
|---|---|---|---|---|---|
| **Ref.** | **Group** | | | | **Q. Num.** |
| | **G1** | **G2** | **G3** | **G4** | |
| *Redelinghuys, Basson & Kruger (2019)* | 1 | 1 | 1 | 0 | 3,8,13 |
| *Brambilla, Umuhoza & Acerbis (2017)* | 0 | 0 | 1 | 1 | 13,19 |
| *Shahin et al. (2019)* | 0 | 0 | 1 | 2 | 13,16,17 |
| *Zimmermann (2017a)* | 0 | 0 | 2 | 0 | 13,14 |
| *Zimmermann (2017b)* | 0 | 1 | 1 | 3 | 11,13,16–18 |
| *Tien et al. (2019)* | 2 | 1 | 0 | 0 | 2,5,6 |
| *Oppermann et al. (2018)* | 0 | 0 | 1 | 0 | 13 |
| *Brucker et al. (2017)* | 1 | 0 | 1 | 0 | 5,13 |
| *Krishnan, Duttagupta & Achuthan (2019)* | 2 | 0 | 2 | 1 | 2,5,13,14,17 |
| *Salibindla (2018)* | 1 | 0 | 0 | 0 | 2 |
| *Nguyen & Baker (2019)* | 1 | 0 | 1 | 0 | 2,13 |
| *Pustchi, Krishnan & Sandhu (2015)* | 4 | 1 | 2 | 0 | 2–6,13,15 |
| *Garg & Garg (2019)* | 1 | 1 | 0 | 2 | 5,11,16,17 |
| *Souppaya, Morello & Scarfone (2017)* | 3 | 3 | 1 | 2 | 2,4,5,8,10,11,13,16,17 |
| *Brenner et al. (2017)* | 0 | 1 | 0 | 0 | 11 |
| *Vassilakis, Panaousis & Mouratidis (2016)* | 2 | 0 | 0 | 0 | 2,4 |
| *Yarygina (2018)* | 1 | 3 | 2 | 2 | 2,6,8,11,13,14,16,17 |
| *Beekman & Porter (2017)* | 2 | 0 | 1 | 0 | 2,5,13 |
| *Syed & Fernandez (2017)* | 3 | 0 | 2 | 0 | 2,4,5,13,15 |
| *Syed & Fernandez (2018)* | 2 | 0 | 3 | 1 | 2,4,13,14,15,16 |
| *Bhattacharya (2019)* | 0 | 3 | 1 | 0 | 6,7,8,13 |
| *Zhang et al. (2017)* | 0 | 2 | 3 | 0 | 6,7,13,14,15 |
| *Zaheer et al. (2019)* | 1 | 0 | 1 | 0 | 5,13 |
| *Walsh & Manferdelli (2017)* | 0 | 0 | 1 | 0 | 13 |
| *Torkura, Sukmana & Meinel (2017)* | 1 | 2 | 2 | 2 | 2,6,11,13,14,16,17 |
| *Clancy, McGwier & Chen (2019)* | 0 | 0 | 2 | 0 | 13,14 |
| *Cerny, Sedlisky & Donahoo (2018)* | 0 | 0 | 2 | 2 | 13,14,18,19 |
| *Tourani et al. (2019)* | 1 | 3 | 3 | 0 | 3,6,7,8,13,14,15 |
| *Chen, Huang & Chen (2019)* | 1 | 1 | 1 | 0 | 5,8,13 |
| *Anisetti et al. (2019)* | 0 | 0 | 1 | 2 | 15,16,17 |
| *Leite et al. (2019)* | 0 | 0 | 2 | 2 | 13,14,16,17 |
| *Suneja, Kanso & Isci (2019)* | 0 | 2 | 1 | 0 | 8,11,13 |
| *Schlossnagle (2018)* | 0 | 0 | 1 | 2 | 13,16,17 |
| *Schlossnagle (2017)* | 0 | 0 | 1 | 2 | 13,16,17 |
| *Guija & Siddiqui (2018)* | 4 | 0 | 1 | 0 | 2,3,4,5,13 |
| *Esparrachiari, Reilly & Rentz (2018)* | 2 | 0 | 0 | 0 | 2,5 |
| *Gupta, Venkatachalapathy & Jeberla (2019)* | 0 | 0 | 0 | 2 | 16,17 |
| *Troiano et al. (2019)* | 2 | 0 | 2 | 0 | 2,3,13,14 |
| *Tchoubraev & Wiczynski (2015)* | 0 | 1 | 0 | 0 | 11 |
| *Sun, Nanda & Jaeger (2015)* | 2 | 5 | 2 | 0 | 3,5,6,7,8,10,11,13,14 |

| (continued) | | | | | |
|---|---|---|---|---|---|
| **Ref.** | **Group** | | | | **Q. Num.** |
| | **G1** | **G2** | **G3** | **G4** | |
| *Callegati et al. (2016)* | 2 | 4 | 1 | 0 | 3,4,6,7,8,11,13 |
| *Thanh et al. (2016)* | 2 | 3 | 1 | 2 | 2,5,6,7,8,13,16,17 |
| *Ahmadvand & Ibrahim (2016)* | 0 | 0 | 1 | 0 | 13 |
| *Kelbert et al. (2017)* | 0 | 0 | 2 | 0 | 13,14 |
| *George & Mahmoud (2017)* | 2 | 0 | 1 | 0 | 2,5,13 |
| *Esposito et al. (2017)* | 1 | 0 | 1 | 0 | 2,13 |
| *Torkura et al. (2017)* | 2 | 3 | 1 | 2 | 4,5,8,11,12,14,16,17 |
| *Yarygina & Bagge (2018)* | 1 | 0 | 1 | 2 | 4,13,17,18 |
| *Trihinas et al. (2018)* | 0 | 0 | 1 | 1 | 13,16 |
| *Bánáti et al. (2018)* | 3 | 0 | 2 | 0 | 2,3,5,13,14 |
| *Pahl, Aubet & Liebald (2018)* | 1 | 0 | 1 | 0 | 5,13 |
| *Diekmann et al. (2018)* | 0 | 0 | 1 | 1 | 13,17 |
| *Trihinas, Tryfonos & Dikaiakos (2016)* | 0 | 0 | 2 | 1 | 13,14,16 |
| *Nehme et al. (2019)* | 0 | 0 | 1 | 2 | 13,16,17 |
| *Torkura, Sukmana & Kayem (2018)* | 1 | 1 | 1 | 0 | 4,8,13 |
| *Gerking & Schubert (2019)* | 1 | 0 | 1 | 0 | 5,13 |
| *Bogner et al. (2019)* | 0 | 0 | 1 | 4 | 13,16,17,18,19 |
| *Petrovska, Memeti & Imeri (2019)* | 2 | 0 | 1 | 0 | 2,5,14 |
| *Osman et al. (2019)* | 1 | 2 | 2 | 0 | 5,6,7,13,14 |
| *Preuveneers & Joosen (2019)* | 1 | 0 | 2 | 1 | 5,13,14,16 |
| *Chen (2019)* | 1 | 1 | 1 | 0 | 4,8,13 |
| *Wu et al. (2019)* | 1 | 1 | 1 | 0 | 2,11,14 |
| *Li et al. (2019b)* | 1 | 1 | 0 | 0 | 2,11 |
| *Ruan et al. (2019)* | 0 | 0 | 0 | 0 | |
| *Mansfield-Devine (2018)* | 1 | 1 | 1 | 1 | 4,8,13,16 |
| *Baboi, Iftene & Gfu (2019)* | 0 | 0 | 1 | 0 | 13 |
| *Trubiani et al. (2018)* | 3 | 1 | 0 | 1 | 2,4,5,8,16 |
| *Krämer, Frese & Kuijper (2019)* | 1 | 0 | 1 | 0 | 5,13 |
| *Varghese & Buyya (2018)* | 0 | 0 | 1 | 0 | 13 |
| *Elsayed & Zulkernine (2019)* | 1 | 1 | 1 | 0 | 4,8,13 |
| *Reyna et al. (2018)* | 1 | 0 | 1 | 0 | 2,13 |
| *Vaquero et al. (2019)* | 0 | 0 | 1 | 0 | 13 |
| *Kochovski et al. (2019)* | 0 | 0 | 2 | 0 | 13,14 |
| *Lwakatare et al. (2019)* | 0 | 2 | 0 | 2 | 8,11,16,17 |
| *Avritzer et al. (2020)* | 1 | 2 | 0 | 0 | 2,6,8 |
| *Nagothu et al. (2018)* | 1 | 1 | 0 | 0 | 2,11 |
| *Baker & Nguyen (2019)* | 3 | 2 | 0 | 0 | 2,4,5,9,11 |
| *Buzachis & Villari (2018)* | 1 | 1 | 0 | 0 | 2,11 |
| *Yuan et al. (2019)* | 0 | 0 | 2 | 0 | 13,14 |
| *Preuveneers & Joosen (2017)* | 1 | 0 | 1 | 0 | 5,13 |

| Ref. | Group | | | | Q. Num. |
|------|-------|---|---|---|---------|
| | G1 | G2 | G3 | G4 | |
| *(continued)* | | | | | |
| *Taha, Talhi & Ould-Slimanec (2019)* | 1 | 1 | 1 | 0 | 2,8,13 |
| *He & Yang (2017)* | 1 | 1 | 0 | 1 | 3,11,16 |
| *Sultan, Ahmad & Dimitriou (2019)* | 3 | 2 | 2 | 1 | 2,4,5,8,11,13,14,16 |
| *De Donno et al. (2019)* | 1 | 3 | 2 | 0 | 2,8,9,11,13,14 |
| *Ghayyur et al. (2018)* | 0 | 2 | 1 | 0 | 8,11,13 |
| *Zhiyi, Shahidehpour & Xuan (2018)* | 0 | 0 | 1 | 0 | 13 |
| *Sim, Barus & Jaya (2019)* | 0 | 0 | 0 | 0 | |
| *Xu, Jin & Kim (2019)* | 2 | 1 | 1 | 0 | 2,3,8,13 |
| *Badii et al. (2019)* | 2 | 2 | 1 | 0 | 3,5,11,12,13 |
| *Yang et al. (2018)* | 1 | 1 | 1 | 0 | 4,8,13 |
| *Di Salle, Gallo & Pompilio (2016)* | 0 | 0 | 1 | 1 | 13,16 |
| *Kang et al. (2018)* | 0 | 0 | 2 | 0 | 13,14 |
| *Brito et al. (2019)* | 1 | 3 | 1 | 0 | 4,8,11,12,13 |
| *Casale et al. (2019)* | 0 | 0 | 2 | 3 | 13,14,16,17,18 |
| *Di Ciccio et al. (2019)* | 0 | 2 | 1 | 1 | 9,10,14,19 |
| *Kathiravelu, Van Roy & Veiga (2019)* | 0 | 1 | 1 | 0 | 11,13 |
| *Łaskawiec, Choraś & Kozik (2019)* | 0 | 1 | 1 | 0 | 11,13 |
| *Leite et al. (2017)* | 0 | 1 | 1 | 0 | 11,13 |
| *Redelinghuys, Basson & Kruger (2019)* | 1 | 1 | 1 | 0 | 3,8,13 |
| *Brambilla, Umuhoza & Acerbis (2017)* | 0 | 0 | 1 | 1 | 13,19 |
| *Shahin et al. (2019)* | 0 | 0 | 1 | 2 | 13,16,17 |
| *Zimmermann (2017a)* | 0 | 0 | 2 | 0 | 13,14 |
| *Zdun, Wittern & Leitner (2019)* | 0 | 1 | 1 | 2 | 11,13,16,19 |

| Ref. | Group | | | | Q. Num. |
|------|-------|---|---|---|---------|
| | G1 | G2 | G3 | G4 | |
| Theoretical and Applicative Publications | | | | | |
| *Ahmadvand et al. (2018)* | 2 | 7 | 3 | 3 | 2,3,6–18 |
| *Forti, Ferrari & Brogi (2020)* | 0 | 0 | 2 | 0 | 13,14 |
| *Díaz-Sánchez et al. (2019)* | 2 | 1 | 1 | 0 | 4,5,11,13 |
| *Han et al. (2019)* | 3 | 1 | 1 | 0 | 2,3,5,9,13 |
| *Paladi, Michalas & Dang (2018)* | 2 | 4 | 2 | 1 | 2,4,6,8,9,12,13,14,17 |
| *Stocker et al. (2018)* | 2 | 2 | 1 | 0 | 2,5,8,12,13 |
| *Andersen et al. (2018)* | 3 | 2 | 2 | 0 | 2,3,4,8,11,13,14 |
| *Andersen et al. (2018)* | 3 | 2 | 2 | 0 | 2,3,4,8,11,13,14 |
| *Li et al. (2018)* | 4 | 5 | 3 | 2 | 2–9,11,13-15,18,19 |
| *Akkermans et al. (2018)* | 1 | 3 | 2 | 0 | 3,6,7,9,13,14 |
| *Nikouei et al. (2019)* | 1 | 1 | 2 | 0 | 5,8,13,14 |
| *Nagendra et al. (2019)* | 4 | 1 | 1 | 0 | 2–6,13 |
| *Wang et al. (2018)* | 2 | 0 | 0 | 1 | 2,3,16 |
| *Basso et al. (2017)* | 1 | 1 | 1 | 0 | 2,9,13 |

*(Continued)*

| Ref. | Group | | | | Q. Num. |
|---|---|---|---|---|---|
| | G1 | G2 | G3 | G4 | |
| Marchal, Cholez & Festor (2018) | 2 | 0 | 2 | 0 | 2,3,13,14 |
| Demoulin et al. (2018) | 3 | 0 | 0 | 0 | 2,3,5 |
| Pahl & Donini (2018) | 1 | 0 | 2 | 1 | 5,13,14,20 |
| Kang, Shin & Kim (2019) | 2 | 1 | 2 | 1 | 3,4,8,13,14,17 |
| Osman, Hanisch & Strufe (2019) | 0 | 0 | 0 | 1 | 17 |
| Xu et al. (2019) | 2 | 1 | 1 | 1 | 2,3,11,13,18 |
| da Silva, de Oliveira Silva & Brito (2019) | 2 | 1 | 0 | 0 | 2,4,9 |
| Jin et al. (2019) | 3 | 1 | 0 | 0 | 2,3,4,12 |
| Wen et al. (2019) | 2 | 2 | 0 | 0 | 3,4,8,12 |
| Callegati et al. (2018) | 3 | 1 | 2 | 0 | 2,4,5,8,13,14 |
| Jander, Braubach & Pokahr (2018) | 2 | 1 | 1 | 2 | 2,3,11,13,16,20 |
| Jander, Braubach & Pokahr (2019) | 2 | 1 | 1 | 1 | 2,3,11,13,20 |
| Surantha & Ivan (2019) | 3 | 1 | 1 | 1 | 3–5,10,13,20 |
| Hole (2016) | 4 | 2 | 3 | 1 | 2,3,4,5,8,11,13-15,16 |
| Ravichandran, Taylor & Waterhouse (2016) | 4 | 1 | 2 | 2 | 2–5,8,13,14,16,17 |
| Otterstad & Yarygina (2017) | 2 | 3 | 2 | 1 | 2,3,6,7,8,13,14,17 |
| Yarygina & Otterstad (2018) | 1 | 3 | 2 | 0 | 3,6,7,8,13,14 |
| Luo, Ren & Zhang (2018) | 0 | 1 | 3 | 3 | 8,13–18 |
| Camilli et al. (2017) | 2 | 1 | 3 | 4 | 2,3,8,13–19 |
| Nkomo & Coetzee (2019) | 3 | 3 | 3 | 1 | 2,3,5,8,9,11,13–15,17 |
| Beheshti et al. (2019) | 0 | 1 | 2 | 0 | 9,13,14 |
| Chidambaram et al. (2019) | 2 | 0 | 1 | 0 | 3,5,13 |
| Jan et al. (2019) | 1 | 3 | 1 | 0 | 5,6,7,8,14 |
| Melis et al. (2018) | 1 | 1 | 2 | 0 | 3,12,13,14 |
| Paschke (2016) | 2 | 0 | 2 | 0 | 2,3,13,14 |
| Prandi et al. (2019) | 0 | 0 | 1 | 0 | 13 |
| Ibrahim, Bozhinoski & Pretschner (2019) | 3 | 0 | 1 | 2 | 2,3,4,13,16,17 |
| Ranjbar et al. (2017) | 2 | 0 | 1 | 0 | 2,3,13 |
| Han et al. (2019) | 3 | 1 | 1 | 0 | 2,3,5,9,13 |
| Paladi, Michalas & Dang (2018) | 2 | 4 | 2 | 1 | 2,4,6,8,9,12–14,17 |
| Stocker et al. (2018) | 2 | 2 | 1 | 0 | 2,5,8,12,13 |
| Andersen et al. (2018) | 3 | 2 | 2 | 0 | 2,3,4,8,11,13,14 |
| Li et al. (2018) | 4 | 5 | 3 | 2 | 2–9,11,13–15,18,19 |
| Akkermans et al. (2018) | 1 | 3 | 2 | 0 | 3,6,7,9,13,14 |
| Nikouei et al. (2019) | 1 | 1 | 2 | 0 | 5,8,13,14 |
| Nagendra et al. (2019) | 0 | 0 | 0 | 0 | |
| Wang et al. (2018) | 2 | 0 | 0 | 1 | 2,3,16 |
| Basso et al. (2017) | 1 | 1 | 1 | 0 | 2,9,13 |
| Marchal, Cholez & Festor (2018) | 2 | 0 | 2 | 0 | 2,3,13,14 |
| Demoulin et al. (2018) | 3 | 0 | 0 | 0 | 2,3,5 |

| | Group | | | | Q. Num. |
|---|---|---|---|---|---|
| **(continued)** | | | | | |
| **Ref.** | | | | | |
| | **G1** | **G2** | **G3** | **G4** | |
| *Pahl & Donini (2018)* | 1 | 0 | 2 | 1 | 5,13,14,20 |
| *Kang, Shin & Kim (2019)* | 2 | 1 | 2 | 1 | 3,4,8,13,14,17 |
| *Osman, Hanisch & Strufe (2019)* | 0 | 0 | 0 | 1 | 17 |
| *Xu et al. (2019)* | 2 | 1 | 1 | 1 | 2,3,11,13,18 |
| *da Silva, de Oliveira Silva & Brito (2019)* | 2 | 1 | 0 | 0 | 2,4,9 |
| *Jin et al. (2019)* | 3 | 1 | 0 | 0 | 2,3,4,12 |
| *Wen et al. (2019)* | 2 | 2 | 0 | 0 | 3,4,8,12 |
| *Abidi et al. (2019)* | 3 | 1 | 2 | 0 | 3,4,5,8,13,14 |
| *Callegati et al. (2018)* | 3 | 1 | 2 | 0 | 2,4,5,8,13,14 |
| *Thramboulidis, Vachtsevanou & Kontou (2019)* | 1 | 1 | 1 | 0 | 5,8,13 |
| *Jander, Braubach & Pokahr (2018)* | 2 | 1 | 1 | 2 | 2,3,11,13,16,20 |
| *Jander, Braubach & Pokahr, 2019* | 2 | 1 | 1 | 1 | 2,3,11,13,20 |
| *Surantha & Ivan (2019)* | 3 | 1 | 1 | 1 | 3,4,5,10,13,20 |
| *Ciavotta et al. (2017)* | 0 | 1 | 1 | 1 | 11,13,17 |
| *Díaz-Sánchez et al. (2019)* | 2 | 1 | 1 | 0 | 4,5,11,13 |
| *Hole (2016)* | 4 | 2 | 3 | 1 | 2–5,8,11,13–16 |
| *Ravichandran, Taylor & Waterhouse (2016)* | 4 | 1 | 2 | 2 | 2–5,8,13,14,16,17 |
| *Otterstad & Yarygina (2017)* | 2 | 3 | 2 | 1 | 2,3,6,7,8,13,14,17 |
| *Yarygina & Otterstad (2018)* | 1 | 3 | 2 | 0 | 3,6,7,8,13,14 |
| *Luo, Ren & Zhang (2018)* | 0 | 1 | 3 | 3 | 8,13,14,15,16,17,18 |
| *Camilli et al. (2017)* | 2 | 1 | 3 | 4 | 2,3,8,13,14,15,16,17,18,19 |
| *Ahmadvand et al. (2018)* | 2 | 7 | 3 | 3 | 2,3,6–18 |
| *Nkomo & Coetzee (2019)* | 3 | 3 | 3 | 1 | 2,3,5,8,9,11,13–15,17 |
| *Beheshti et al. (2019)* | 0 | 1 | 2 | 0 | 9,13,14 |
| *Chidambaram et al. (2019)* | 2 | 0 | 1 | 0 | 3,5,13 |
| *Jan et al. (2019)* | 1 | 3 | 1 | 0 | 5,6,7,8,14 |
| *Melis et al. (2018)* | 1 | 1 | 2 | 0 | 3,12,13,14 |
| *Paschke (2016)* | 2 | 0 | 2 | 0 | 2,3,13,14 |
| *Prandi et al. (2019)* | 0 | 0 | 1 | 0 | 13 |
| *Ibrahim, Bozhinoski & Pretschner (2019)* | 3 | 0 | 1 | 2 | 2,3,4,13,16,17 |
| *Ranjbar et al. (2017)* | 2 | 0 | 1 | 0 | 2,3,13 |
| *Ranawaka et al. (2020)* | 2 | 1 | 1 | 0 | 2,3,11,13 |
| *Du et al. (2020)* | 2 | 1 | 2 | 0 | 2,3,11,13,15 |
| *Haque, Iwaya & Babar (2020)* | 4 | 1 | 1 | 2 | 2,3,4,5,11,13,16,17 |
| *Avritzer et al. (2020)* | 3 | 4 | 1 | 4 | 2–4,6,7,9,11,13,16–19 |
| *Alaluna et al. (2020)* | 3 | 1 | 0 | 0 | 2,3,4,11 |
| *Falah et al. (2020)* | 1 | 1 | 0 | 0 | 2,6 |
| *Truong & Klein (2020)* | 2 | 1 | 1 | 1 | 2,4,11,13,16 |
| *Nikolakis et al. (2020)* | 3 | 1 | 2 | 0 | 2,3,4,11,13,14 |
| *Kumar & Goyal (2020)* | 2 | 4 | 2 | 3 | 2,4,6,7,8,11,13,14,16,17,18 |

(Continued)

| (continued) | | | | | |
| Ref. | Group | | | | Q. Num. |
| | G1 | G2 | G3 | G4 | |
| *Janjua et al. (2020)* | 3 | 1 | 1 | 1 | 2,3,4,11,13,20 |
| *Hahn, Davidson & Bardas (2020)* | 3 | 4 | 1 | 1 | 2,3,4,8,9,10,11,13,16 |
| *Cheruvu et al. (2020)* | 2 | 0 | 0 | 0 | 2,3 |
| *Lakhan & Li (2020)* | 0 | 1 | 1 | 0 | 11,13 |
| *Javed et al. (2020)* | 2 | 0 | 1 | 0 | 3,4,13 |
| *Lou et al. (2020)* | 4 | 4 | 0 | 0 | 2,3,4,5,6,7,10,11 |
| *Maati & Saidouni (2020)* | 4 | 0 | 0 | 0 | 2,3,4,5 |
| *Lu et al. (2021)* | 2 | 1 | 0 | 2 | 2,3,11,18,19 |
| *Copei, Wickert & Zündorf (2020)* | 3 | 1 | 1 | 1 | 2,4,5,11,13,20 |
| *Ranawaka et al. (2020)* | 2 | 1 | 1 | 0 | 2,3,11,13 |

## Funding
Fabrizio Montesi was supported by Villum Fonden, grant no. 29518, and by Independent Research Fund Denmark, grant no. 0135-00219. The funders had no role in study design, data collection and analysis, decision to publish, or preparation of the manuscript.

## Grant Disclosures
The following grant information was disclosed by the authors:
Villum Fonden: 29518.
Independent Research Fund Denmark: 0135-00219.

## Competing Interests
The authors declare that they have no competing interests.

## Author Contributions
- Davide Berardi performed the experiments, performed the computation work, prepared figures and/or tables, and approved the final draft.
- Saverio Giallorenzo conceived and designed the experiments, prepared figures and/or tables, authored or reviewed drafts of the paper, and approved the final draft.
- Jacopo Mauro conceived and designed the experiments, authored or reviewed drafts of the paper, and approved the final draft.
- Andrea Melis conceived and designed the experiments, performed the experiments, analyzed the data, performed the computation work, prepared figures and/or tables, and approved the final draft.
- Fabrizio Montesi conceived and designed the experiments, authored or reviewed drafts of the paper, and approved the final draft.

- Marco Prandini conceived and designed the experiments, prepared figures and/or tables, authored or reviewed drafts of the paper, and approved the final draft.

## Data Availability

The data is available on Zenodo: Davide Berardi, Saverio Giallorenzo, Jacopo Mauro, Andrea Melis, Fabrizio Montesi, & Marco Prandini. (2021). Microservice Security: A Systematic Literature Review, dataset (1.0) [Data set]. Zenodo. https://doi.org/10.5281/zenodo.5513580.

## Supplemental Information

Supplemental information for this article can be found online at http://dx.doi.org/10.7717/peerj-cs.779#supplemental-information.

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
