# Peer review of "Microservice security: a systematic literature review"

_PeerJ Computer Science, doi:10.7717/peerj-cs.779_

## Round 0.1 · original submission · Major Revisions

Dear Authors,

Based on the comments received from the reviewers and my own observations, I recommend major revisions for the article.

Reviewer 1 ·

Basic reporting

Microservices is an emerging paradigm for developing distributed systems. With their widespread adoption, more and more work investigated the relation between microservices and security. In this work, the authors conduct a systematic review of the field, gathering 290 relevant publications. The paper is well written and organized. However, it needs significant updations before it can b e accepted

Experimental design

Good

Validity of the findings

Satisfactory.

Additional comments

1. In the introduction, the authors can draw a table showing the difference between the current survey and existing surveys.
2. Some of the recent papers related to security such as the following can be discussed in the paper: "Iwendi, C., Jalil, Z., Javed, A. R., Reddy, T., Kaluri, R., Srivastava, G., & Jo, O. (2020). Keysplitwatermark: Zero watermarking algorithm for software protection against cyber-attacks. IEEE Access, 8, 72650-72660., Bhardwaj, A., Shah, S. B. H., Shankar, A., Alazab, M., Kumar, M., & Gadekallu, T. R. (2020). Penetration testing framework for smart contract Blockchain. Peer-to-Peer Networking and Applications, 1-16."
3. List out the main contributions of the survey.
4. The authors can add a section, lessons learnt and future directions to pave the way for researchers interested to work on this topic.

Reviewer 2 ·

Basic reporting

Well written, well structured.

One important issue with the use of references. (see later)

Experimental design

According to the standards of a literature survey.

However, I would like to see an explicit motivation on the exclusion of the grey literature.

Validity of the findings

The authors conclude with a convincing call for action to help the community make steps forward.

Additional comments

Summary.

This paper reports on a systematic literature survey on microservice security. They analysed 290 relevant publications (published since 2014) in a quantitative and qualitative way. They complement this analysis by a thorough knowledge about the field and as such identify open areas where future research is welcome. They conclude with a convincing call for action to help the community make steps forward.

Points in favour

* The authors conducted the literature survey according to the guidelines for doing so. The data is publicly available as I would expect from a publication appearing in PeerJ On top of that they conducted a meta analysis with clusters of authors and a word-net of abstracts which was a cherry on the pie as far as quantitative analysis goes.

* I was suspicious with the qualitative analysis driven by 20 (twenty!) research questions. However, as later explained what is actually happening is that these 20 questions serve as markers to classify whether these papers cover a certain topic or not. This allows for the correlation matrix (Table 2) which was again a nice complement.

* The field is quite diverse which is a good sign for a literature survey. The publication outlets and the research communities illustrate quite well that a systematic literature survey like this one is due.

* Throughout the paper I feel a sense of credibility: the authors know what they are talking about and are able to link and integrate findings from a very diverse field. The discussion and conclusion section formulated a s a series of open challenges is therefore very convincing. I especially appreciated the "React & recover Techniques" and the "DevSecOps"



Suggestions for improvement
(All below are considered minor. I trust that the authors will do their best to accommodate these suggestions and if they do not I can live with the paper)


* They way to authors list papers in the bibliography is very annoying to look up references. In the end I resorted to a search on the digital PDF document while reading the article in paper form.

Citations start with the family name of the first author followed by the year (e.g. Casale et al.[2016]) However in the bibliography the first name comes first, family name comes second followed by all other author names; title, misc and the year at the very end of the entry. (i.e. Giuliano Casale, Cristina Chesta, ... Cloud Forward, pages 34–42, 2016). The alphabetical sorting of the references is thus hard to infer and scanning the list is counterintuitive. Note that some authors have multiple first names (e.f. Mohammad Bany Taha) which makes it even more difficult to discern the alphabetical sorting of the references.
The authors also split the reference section in two subsection (References + Publications from the Dataset). So scanning from the end of the bibliography was awkward. (Looking for the entry "Wuyts 538 et al. [2018]" I started from the back and did not find it on the first attempt. Only the PDF search revealed what was happening)


* I would like to authors to explicitly motivate why they did not include grey literature in their survey. Especially in a field like microservices which is driven by practitioners this may provide complementary perspective. Perhaps there would be blog posts on "React & recover Techniques" and the "DevSecOps" which makes the call for action even more worthwhile.

* "Methodology." Don't use the term "methodology" for what is a method, it is inflation of words. The postfix -OLOGY stands for "study of". (i.e. biology = the study of the living organisms; psychology = is the study of the human mind; geology = is the study of the earth). Thus methodology is "the study of the methods".

* lines 101-102 "We analysed these publications to collect statistical and objective answers". There is no such thing as an "objective" answer. The very fact that you classified papers already implies a subjective interpretation. I would suggest to replace "objective" by this by "transparant".

* Line 360-361 "This is reflected by the sharp increase in the number of publications since 2014." I am not convinced by this argument. The overall number of publications in computers science is steadily increasing, so the absolute number of papers does not say much.

·

Basic reporting

The authors present a systematic literature review on microservices’ security, in a well-written manuscript. In addition, the topic is certainly timely and relevant, and one such review can provide various benefits to both researchers and practitioners working on the topic. At the same time, the submitted manuscript must be revised to address various limitations it is currently suffering from, before it can be considered for publication in an international journal. I am listing such limitations in the following sections, together with suggestions hopefully helping the authors in improving their manuscript.

Experimental design

As for the search for studies, some concerns should be clarified by the authors. Firstly, why did they choose to restrict their focus to “white literature” only? It is known that industry is heavily investing on microservices, with quite many solutions (also for security) being proposed and posted in industry-driven outlets (whitepapers, blog posts, YouTube channels, etc.). I am not saying that the authors should include “grey literature” as well, but they should at least clarify why they decided to keep it out from their study. (The authors actually notice this in their conclusions, hence raising the point of why they did not consider grey literature from the beginning)

In addition, I am not sure on the “repeatability” of one of the exclusion criteria, viz., “We also excluded cases in which the work tangentially mentioned the satisfaction of some security aspects, without detailing the design/development of the security technologies to accomplish them”. Whilst all other exclusion criteria are objective, this criteria seem to potentially be threaten by observer bias (as an observer may a manuscript to satisfy this conditions and hence get excluded, while another may consider the treating of security “not so tangent”, so to say). The authors should clarify how they limited/avoided possible observer biases when evaluating this criteria, e.g., in a “Threats to Validity” section.

Finally, the authors should conclude this section by listing all selected studies (e.g., in a table). This would help keeping the authors review self-contained, hence helping readers in better understanding their results.

As for the research questions, the authors state that they adopted 20 dichotomous questions with the goal of favoring precision and objectiveness. It is however not clear whether/how they ensured such precision and objectiveness. The marking of a publication as “yes” or “no” for a research question seems indeed to be subject to observer biases and errors. How did the authors avoid/limit this? This should be discussed in a “Threats to Validity” section.

Validity of the findings

The presentation of results can and must be improved, by also expanding their discussion. For instance, the authors should consider presenting the publication outlets (viz., conferences and journals) as plots, so that readers can visually observe them (rather than finding a flat list of names). Much better it would be to cluster venues according to some criteria and to show aggregated results. For instance, the authors could expand their discussion on the communities where microservices’ security is most discussed (with some graphical support).

Also, I am not sure on whether what the authors call “qualitative results” actually provide “qualitative” information. For instance, the distribution of types of publication is still a “quantitative” information, with which the authors partition the different contributions in the field (here the authors use the word “survey” to denote “reviews”). This type of information is typically present in systematic literature reviews, and it is usually classified as a “quantitative result”.

The same holds for the other aspects discussed in Sects. 5.2.2-5. The authors state “how many” publications were marked as “yes” for the research questions pertaining to each aspect, hence providing “quantitative information” on the distribution for such aspects. To make the things more “qualitative” the authors should better enter into the details of "what" is discussed in each publication and "how". Whilst the “what” can be easily shown with tables showing the authors’ classification of considered works (e.g., with rows associated with works, columns with research questions, and checkmarks placed in cells to denote that a research question is treated in a research work), the “how” requires the authors to expand their discussion by discussing how (clusters of) works discuss/tackle the research questions.

As a minor comment, in Section 5.2.6 the authors state that they “were surprised to find many citations to blockchain technologies (as reported above) as well as the lack of more and more mainstream technologies like service mesh and serverless”. I am not sure on whether “service mesh” are more mainstream than “blockchains”. The authors should consider rewording this, or at least cite a reference showing the higher recognition/usage of “service mesh” if compared with “blockchains”.

In their concluding remarks (Sect. 6), the authors draw some concluding remarks on the open challenges that emerged from their study. A reader however misses the links between the results presented in Sect. 5 and the open challenges/research directions listed in Sect. 6. The authors should try to make such links more explicit. For instance, they could introduce open challenges (e.g., in a “highlighting box”) in Sect. 5, immediately after the discussion highlighting the need/openness of such challenge. They could then retake/recap the open challenges (as they currently do) in Sect. 6.

Last, but not least, systematic literature reviews, and systematic studies in general, are known to be prone to possible threats to their validity (like those I tried to highlight in my former comments above). The authors should discuss how they mitigated/avoided possible threats to the validity of their study in a devoted section.

Additional comments

Other comments follow:

[ “survey” vs. “review” ]
I am not sure on whether the word “survey” used by the authors in the manuscript’s title and throughout the text is correct. Snyder’s guidelines, used by the authors to design their research, speak about “systematic literature reviews”. I would hence recommend the authors to revise their wording in “systematic literature review”, both in the title and all over the manuscript. Such wording is that most commonly associated with the type of studies like that presented by the authors in this manuscript, hence helping potential readers to better grab the type of content they would find in this manuscript.

[ Related Work ]
The related work discussion should be expanded to include other relevant related reviews on “microservices & security”, e.g., that on “microservices security smells” by Ponce et al. (https://arxiv.org/abs/2104.13303) .

[ References ]
The authors should cite all papers they considered in their literature review, to give credit to the authors who published such papers.

[ Appendix ] 
The authors present selected studies and their classification in a supplemental appendix. As noticed in my above comments, such information is not supplemental, but crucial for readers to understand which papers were considered and how they were classified to answer to the authors’ research questions. The information in the appendix should hence be included in the main text of the manuscript.

*** other comments ***
- “Alas” -> “Unfortunately” or “At the same time”?
- p5/28: “within the” -> “up to the”

---

## Round 0.2 · accepted · Accept

I recommend to accept the manuscript since all issues raised by the reviewers have been fully addressed.

Reviewer 2 ·

Basic reporting

No remarks here, 2nd review

Experimental design

No remarks here, 2nd review

Validity of the findings

No remarks here, 2nd review

Additional comments

The authors took into account all suggestions from the reviewers and complied as best as they could

·

Basic reporting

The authors have thoroughly addressed all my former comments, significantly improving their paper. I would hence recommend accepting the paper in its current form.

Experimental design

no comment

Validity of the findings

no comment

Additional comments

no comment